# Deficiency in class III PI3-kinase confers postnatal lethality with IBD-like features in zebrafish

Shaoyang Zhao[1,2,3], Jianhong Xia[2,3], Xiuhua Wu[2,3], Leilei Zhang[2,3], Pengtao Wang[2,3], Haiyun Wang[2,3], Heying Li[2], Xiaoshan Wang [2], Yan Chen [2], Jean Agnetti[2], Yinxiong Li [2], Duanqing Pei[2,3] & Xiaodong Shu[2,3]

The class III PI3-kinase (PIK3C3) is an enzyme responsible for the generation of phosphatidylinositol 3-phosphate (PI3P), a critical component of vesicular membrane. Here, we report that PIK3C3 deficiency in zebrafish results in intestinal injury and inflammation. In *pik3c3* mutants, gut tube forms but fails to be maintained. Gene expression analysis reveals that barrier-function-related inflammatory bowel disease (IBD) susceptibility genes (*e-cadherin*, *hnf4a*, *ttc7a*) are suppressed, while inflammatory response genes are stimulated in the mutants. Histological analysis shows neutrophil infiltration into mutant intestinal epithelium and the clearance of gut microbiota. Yet, gut microorganisms appear dispensable as mutants cultured under germ-free condition have similar intestinal defects. Mechanistically, we show that PIK3C3 deficiency suppresses the formation of PI3P and disrupts the polarized distribution of cell-junction proteins in intestinal epithelial cells. These results not only reveal a role of PIK3C3 in gut homeostasis, but also provide a zebrafish IBD model.

[1] School of Life Sciences, University of Science and Technology of China, 230027 Hefei, Anhui, China. [2] CAS Key Laboratory of Regenerative Biology, Guangzhou Institute of Biomedicine and Health-Guangzhou Medical University Joint School of Biological Sciences, South China Institute for Stem Cell Biology and Regenerative Medicine, Guangzhou Institutes of Biomedicine and Health, Chinese Academy of Sciences, 510530 Guangzhou, China. [3] Guangdong Provincial Key Laboratory of Stem Cell and Regenerative Medicine, South China Institute for Stem Cell Biology and Regenerative Medicine, Guangzhou Institutes of Biomedicine and Health, Chinese Academy of Sciences, 510530 Guangzhou, China. These authors contributed equally: Shaoyang Zhao, Jianhong Xia. Correspondence and requests for materials should be addressed to D.P. (email: pei_duanqing@gibh.ac.cn) or to X.S. (email: shu_xiaodong@gibh.ac.cn)

The integrity of intestinal epithelium is established and maintained by cell junctions between epithelial cells and it is essential for the barrier function against gut microbes. Loss of this integrity is a hallmark of inflammatory bowel disease (IBD), including Crohn's disease (CD) and ulcerative colitis (UC)[1,2]. Genetic studies in animal models have identified genes involved in intestinal epithelial homeostasis. For example, intestinal specific knockout of adherens junction protein E-Cadherin (encoded by the *Cdh1* gene) in mice disrupts the barrier function of intestine and induces epithelial cell death and defective bacterial defensing[3,4]. Hnf4a (hepatocyte nuclear factor 4 alpha) is an epithelial transcription factor involved in the development of liver, intestine, and other tissues. Conditional knockout of this gene in the intestine of adult mice induces down-regulation of the tight junction protein zonula occludens 1 protein (ZO-1), cytoplasmic mis-localization of E-Cadherin and destabilization of epithelial cell–cell junctions[5]. Interestingly, both *CDH1* and *HNF4A* were identified as susceptibility loci in UC patients by genome-wide association studies (GWAS)[6]. Furthermore, disease-associated SNP that results in the cytoplasmic accumulation of E-Cadherin was reported in CD patients[7]. More recently, mutations in tetratricopeptide repeat domain-7A (TTC7A) gene were identified from patients of very early onset IBD (VEO-IBD) or multiple intestinal atresia (MIA), which result in reduced cell adhesion, elevated apoptosis, defective epithelial barrier and polarity[8,9]. In addition to barrier dysfunction, mucosal immunity and intestinal microbiota also play important roles in IBD. Among the 200+ independent genetic risk loci for IBD identified by GWAS[10,11], there are loci that have nearby genes involved in bacterial sensing and autophagy, such as nucleotide-binding oligomerization domain containing 2 (NOD2), autophagy-related 16 like 1 (ATG16L1) and immunity-related GTPase M (IRGM) or inflammatory response, such as interleukin 23 receptor (IL23R), caspase recruitment domain family member 9 (CARD9), matrix metallopeptidase 9 (MMP9), tumor necrosis factor (TNF) and interleukin 1 beta (IL1B) and they are suggested to be causal genetic variants[12–14]. However, the functional relevance for most of the other loci and their associated candidate genes remains to be investigated.

Zebrafish have recently emerged as animal models for human diseases including IBD. The functions of many inflammatory response genes appear conserved between zebrafish and mammal. For example, NOD 1 and 2 are involved in bacterial sensing in mammals and mutation in NOD1 is associated with UC, while NOD2 defect is a high risk for CD. Suppression of these genes in zebrafish reduces the innate immunity against infection by *Salmonella enterica* and increases bacterial burden in infected embryos[15]. myeloid differentiation primary response 88 (MYD88) and IL1B also have similar function in innate immune response in zebrafish and mammals[16,17]. Thus, zebrafish could be used as a platform to evaluate the functional relevance of human GWAS loci. For example, macrophage stimulating 1 (MST1) is a GWAS candidate gene for IBD and disruption of this gene leads to spontaneous intestinal inflammation, increased susceptibility to epithelial damage and prolonged pro-inflammatory response in adult zebrafish[18]. Ubiquitin like with PHD and ring finger domains 1 (UHRF1) and DNA methyltransferase 1 (DNMT1) are regulators of DNA methylation and SNPs near these loci have been identified from IBD patients. Zebrafish *uhrf1* and *dnmt1* mutants show elevated TNFα expression, barrier dysfunction and chronic inflammation in gut[19]. In addition, unbiased genetic screen in zebrafish has identified novel genes involved in intestine function. For example, deficiency of CDP–diacylglycerol–inositol 3-phosphatidyltransferase (CDIPT), an enzyme responsible for the de novo synthesis of intracellular phosphatidylinositol (PI),

induces ER stress, disrupted epithelial structure, and enhanced intestinal inflammation[20].

PIK3C3 (also known as Vps34) is a class III phosphatidylinositol-3-kinase that catalyzes the production of phosphatidylinositol 3-phosphate (PI3P) from PI. Biochemical studies have identified two PIK3C3-containing protein complexes from mammalian cells. The complex I contains PIK3C3/PIK3R4/BECN1/ATG14 and it is involved in autophagy pathway. The complex II consists of PIK3C/PIK3R4/BECN1/UVRAG and it regulates endocytic trafficking[21]. Genetic studies in mice have revealed diverse in vivo functions of PIK3C3 in different tissues. For example, global knockout of PIK3C3 leads to gastrulation defects and early embryonic lethality[22]. Sensory neuron-specific deletion of PIK3C3 induces aggregation of ubiquitinated proteins, accumulation of large vacuoles and neuron degeneration, which is attributed to defect in the endo-lysosomal pathway[23]. Rod-specific ablation of PIK3C3 disrupts the light-induced PI3P production and endosomal trafficking, which leads to progressive degeneration of these cells[24]. Muscle-specific deletion of PIK3C3 induces muscular dystrophy with disrupted intracellular vesicular trafficking[25]. Podocyte-specific deletion of PIK3C3 induces podocyte degeneration and glomerulosclerosis, which also appears to be due to defective endo-lysosomal trafficking[26,27]. Gene ablation in T cells reveals that PIK3C3 regulates the recycling of IL-7Rα and survival of T lymphocytes, which appears autophagy-independent[28]. On the other hand, other studies indicate that the PIK3C3-regulated autophagy is required for the homeostasis of T cells[29,30]. Liver or heart specific deletion of PIK3C3 results in hepatomegaly or cardiomegaly and abnormal autophagy is at least partially responsible for these defects[31]. The potential functions of PI3KC3 in innate immunity or gut homeostasis and pathogenesis remain to be uncovered.

We report here that PIK3C3 deficiency in zebrafish suppresses PI3P synthesis in IECs, induces epithelial damage and pro-inflammatory response in the intestine. The inflammation still occurs in the absence of gut microorganisms and appears mediated by neutrophils. We propose that the zebrafish *pik3c3* mutant may serve as a disease model for mechanistic studies of epithelial damage-induced IBD and a platform for developing and evaluating potential therapeutic interventions.

## Results

**PIK3C3 deficiency results in digestive tract defects in zebrafish.** To investigate the in vivo function of PIK3C3, we generated PIK3C3 knockout zebrafish lines using CRISPR/Cas9 technology (Supplementary Fig. 1a). We obtained *pik3c3* mutant lines with deletions (−19 bp in mutant allele 1 and −2 bp in mutant allele 2) in the exon 11 of *pik3c3* gene. qPCR analysis (with primer 23F/24R) indicates that the mutant RNAs are present at level similar to WT RNA (Supplementary Fig. 1b). No alternative splicing was detected by RT-PCR in the mutants (Supplementary Fig. 1c) and sequencing analysis of those PCR products confirmed the −19 bp and −2 bp deletion in mutant alleles. Both mutations induce frame-shift and generate premature stop in the accessory domain of PIK3C3 (PI3Ka, a domain suggested to be involved in substrate presentation). The truncated proteins contain the intact C2 domain (a domain involved in lipid binding and membrane targeting of PIK3C3), while miss the catalytic domain (PI3Kc) and are expected to be catalytically inactive (Supplementary Fig. 1a). Heterozygous embryos of these alleles grow normally into adulthood with the Mendelian ratio (Supplementary Fig. 1d) and are fertile. Homozygous mutants appear normal in the first week of development, however, they then gradually show curved body and collapsed gut tube (Fig. 1a), and all die by 10 days post

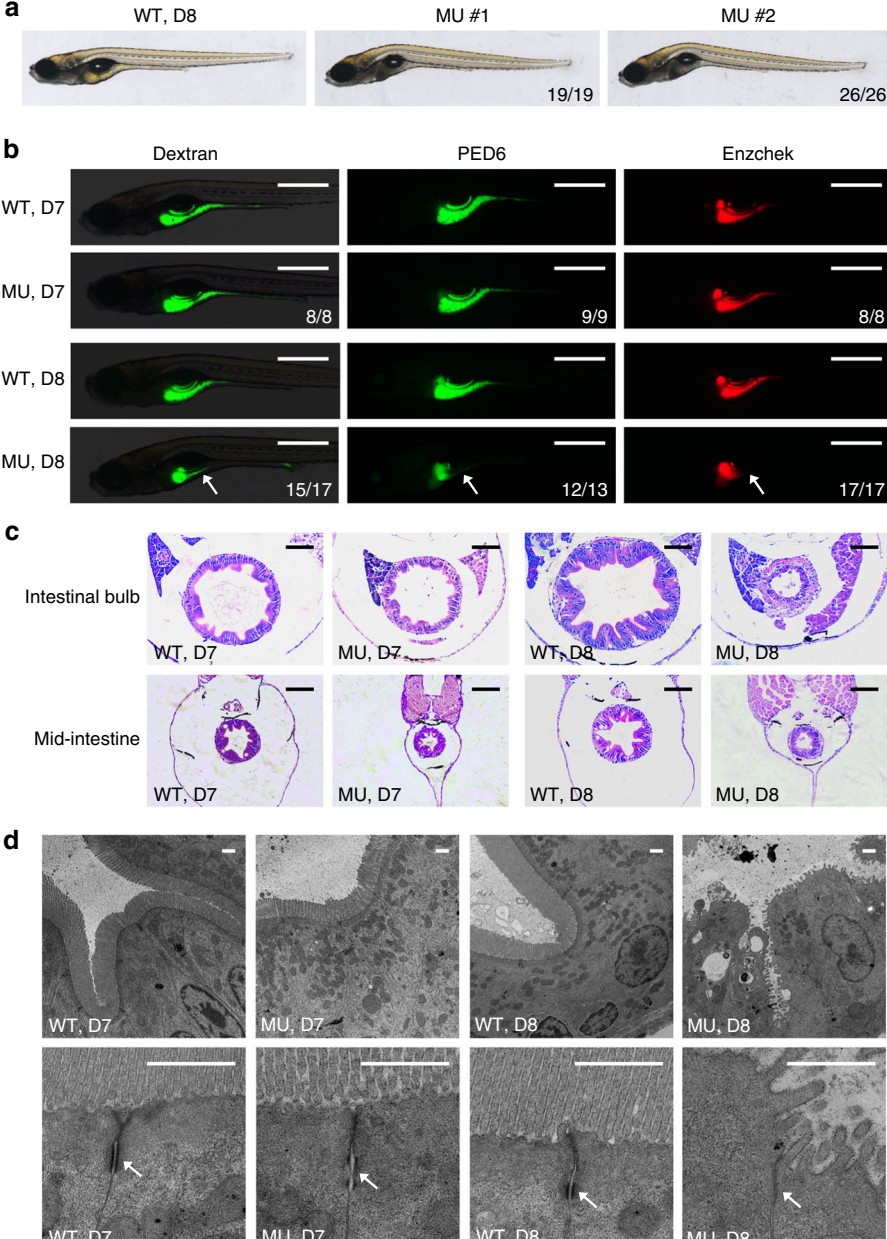

**Fig. 1** Intestinal defects in *pik3c3* mutants. **a** General morphology of WT and *pik3c3* mutants at 8 dpf. **b** In vivo imaging of zebrafish digestive organ with fluorescent reporters. FITC-labeled dextran is used to evaluate the nutrient uptake activity and the quenched fluorescent reporter PED6 or Enzchek is activated only after cleavage by intestinal phospholipase or protease. The digestive functions appear normal in mutants at 7 dpf while they become defective at 8 dpf. Scale bars, 500 μm. Numbers in (**a**, **b**) represent (embryos with the indicated phenotype)/(total embryos analyzed). **c** Hematoxylin–eosin staining of cross sections corresponding to the intestinal bulb or mid-intestine of embryos at 7–8 dpf. Typical intestinal folding is lost and cell shedding can be detected in *pik3c3* mutants at 8 dpf. Scale bars, 50 μm. **d** TEM analyses of IECs. Adherens junctions (arrow) are defective in mutants. Microvilli and cell junctions are severely disrupted in mutants at 8 dpf. Scale bars, 1 μm

fertilization (dpf) (Supplementary Fig. 1d). The mutant phenotype is stably passed through multiple generations and mutant allele 1 and 2 fail to complement with each other, suggesting the specificity of the mutant phenotype. To further analyze the morphological and functional defects of digestive tract in the mutants, we treated embryos with FITC-Dextran (for swallowing activity), PED6 (for lipid processing activity) or Enzchek (for protein processing activity)[32]. These activities appear normal in all embryos at 7 dpf, but they are severely compromised in mutants at 8 dpf (Fig. 1b, Supplementary Fig. 1e). We then performed histological analysis of the digestive tract. Hematoxylin–eosin (H&E) staining of cross sections indicates

that the intestinal bulb and mid-intestine have developed normally in the mutants at 7 dpf. However, those structures appear collapsed at 8 dpf, indicating a failure of maintaining the integrity of intestinal epithelium (Fig. 1c). We further analyzed the structure of intestinal epithelial cells (IECs) by transmission electron microscope (TEM). The IECs are columnar and microvilli are formed while cell junction, such as adhesion belt is defective in mutants at 7 dpf (Fig. 1d). The defects become much severe at 8 dpf as microvilli are disrupted and adherens junctions become barely detectable at this stage (Fig. 1d). Together, these data suggest that PIK3C3 is required for the homeostasis of IECs and digestive function in zebrafish.

**Gene expression changes in the PIK3C3 null embryos.** To uncover PIK3C3 deficiency-induced molecular changes, we performed RNA-Seq analysis. Whole embryonic total RNA samples were prepared from sibling and mutant embryos at 6, 7, and 8 dpf and subjected to sequence with NextSeq 500. Gene ontology (GO) analysis of RNA-Seq data revealed that cellular component related to extracellular matrix and biological processes related to DNA replication and metabolic pathways are down-regulated in the mutants (Fig. 2a and Supplementary Fig. 2). The RNA-Seq data also reveal that many IEC-specific genes, such as apolipoprotein A-1a (*apoa1a*), fatty acid-binding protein 2, intestinal (*fabp2*), villin 1 (*vil1*) and solute carrier family 15 member 1b, also known as pept1 (*slc15a1b*) are down-regulated in the mutants, which is further confirmed by qRT-PCR (Fig. 2a, b). On the other hand, the expressions of liver-specific fatty acid-binding protein 10a (*fabp10a*) and the pancreatic β-cell marker insulin (*ins*) are not suppressed. Interestingly, the expression of exocrine pancreas marker trypsin (*try*) is elevated in the mutants at 8 dpf (Fig. 2b). We then performed in situ hybridization analysis which confirmed that the expression of intestinal *fabp2* or *vil1* is reduced but the pancreatic *try* is not (Fig. 2c). We noticed that intestinal makers, such as *fabp2* and *slc15a1b* are sufficiently induced at 6 dpf but gradually down-regulated afterward in the mutants (Fig. 2b). Together, these results indicate that the down-regulation of intestinal genes is not due to general developmental defects or a failure to specify intestinal cell fate; instead, their expressions fail to be maintained in the mutants.

**PIK3C3 deficiency induces intestinal inflammation.** GO analysis of genes up-regulated in the mutants reveals that biological processes related to inflammatory responses are highly stimulated in the mutants (Supplementary Fig. 3). Heatmap of pro-inflammatory genes indicates that inflammation might occur as early as 6 dpf and it is highly activated at 8 dpf (Fig. 3a). qPCR analysis confirmed the stimulation of *tnfa*, *il1b*, chemokine (C-X-C motif) ligand 8a (*cxcl8a*), leukocyte cell-derived chemotaxin 2 like (*lect2l*), *mmp9* and myeloid-specific peroxidase (*mpx*) (Fig. 3b). This global gene expression analysis reveals the occurrence of inflammatory response but it does not reveal the tissue distribution or the nature of inflammation. Therefore, we took advantages of the zebrafish model to further investigate these questions. At this stage of embryonic development, only innate immunity exists in the zebrafish[33]. So we used the neutrophil (*Tg(mpx:EGFP)*, *mpx* promoter drives EGFP expression) or macrophage (*Tg2(mpeg1:EGFP)*, *mpeg1* promoter drives EGFP expression) lineage labeled transgenic fish line to further investigate the nature of inflammation in the mutants. We found that neutrophils are enriched in the digestive tract of mutants but not wild type siblings at 8 dpf (Fig. 3c, d). Indeed, neutrophil infiltration of intestinal epithelium can be clearly demonstrated in cross sections of mutant gut (Fig. 3e). Neutrophils remain in the mutant gut until the death of embryos, indicating a failure to resolve intestinal inflammation. This is not due to disrupted migratory activity of neutrophils, since both the directional migration of neutrophils in response to tailfin amputation and the reverse migration of neutrophils during inflammation resolution stage appear normal in mutants (Supplementary Fig. 4). Thus, neutrophils per se are not compromised in the mutants. We then examined the distributions of macrophages and found their distributions are comparable between wild type and mutant embryos (Fig. 3f). Together, these results demonstrate that PIK3C3 deficiency induces intestinal accumulation of neutrophils and inflammation in digestive tract.

**PIK3C3 regulates the expressions of intestinal barrier-related genes.** The disruption of epithelial integrity together with the accumulation of neutrophils in the intestine suggests an IBD like phenotype in the PIK3C3 mutants. More than 200 human IBD susceptibility loci have been identified and, currently, verified IBD susceptibility genes can be roughly grouped into three categories: intestinal barrier-related genes, bacterial sensing and autophagy-related genes and inflammatory response genes. We tested whether the expression levels of these genes are changed in the mutant gut. We first micro-dissected digestive tracts from wild type siblings or mutants and total RNAs were isolated from these samples. We performed qRT-PCR analysis to evaluate the protocol and found that the expression level of intestinal gene *fabp2* is enriched more than 60 folds in the dissected gut when compared to whole body extract. Meanwhile, neuronal paired box 2a (*pax2a*), cardic myosin, light chain 7 (cardiac *cmlc2*), and skeletal myosin, heavy polypeptide 1.1 (*myhz1.1*) genes are barely detectable in the isolated gut (Supplementary Fig. 5), indicating the effectiveness of the sample isolation. We then analyzed the expression of IBD candidate genes in the dissected gut and found that the inflammatory response gene, such as *il1b*, *card9*, and *mmp9* are highly stimulated in the mutant gut (Fig. 3g). insulin-like growth factor binding protein 1 (*IGFBP1*) is one of the recently identified candidate genes associated with the prognosis of CD[34]. There are two zebrafish homologs (*igfbp1a* and *igfbp1b*) and *igfbp1a* is one of the most highly expressed genes in the mutants (Fig. 3g). *tnfa* is not significantly induced in the intestine ($p = 0.06$), while it is highly up-regulated in whole embryonic extract (Fig. 3b), suggesting that it is induced in tissues other than digestive tract. We found that bacterial sensing or autophagy pathway-related IBD gene *nod2*, *atg16l1*, and *irgm* are not induced in the mutants, on the other hand, the barrier-related gene *hnf4a* and *ttc7a* are dramatically down-regulated in the mutants. *hp* (*zonulin*), a gene that functions to open up the tight junctions of epithelium[35], is highly stimulated in the mutants (Fig. 3g). Together with our TEM studies (Fig. 1d), it appears that PIK3C3 deficiency disrupts the expression of barrier-related genes and the integrity of intestinal epithelial cell junctions.

**PIK3C3 deficiency results in clearance of gut bacteria.** Intestinal microbiota can play either protective or unfavorable role in IBD. We analyzed whether gut microorganisms are involved in the PIK3C3 deficiency-induced pathogenesis. We raised embryos under either standard or germ-free conditions (Supplementary Fig. 6)[36] and then compared the expressions of inflammatory response genes and lineage markers. We found that under germ-free condition, PIK3C3 deficiency is also able to induce the over-expression of inflammatory genes (*tnfa*, *il1b*, *mmp9*, and *cxcl8a*) and neutrophil marker *mpx* (Fig. 4a). When compared to the standard culture condition, mutants under germ-free condition show higher expression levels of *tnfa*, *il1b*, *mmp9*, and *cxcl8a*. The intestinal epithelial cell marker *fabp2* is reduced to similar level in mutants under both culture conditions (Fig. 4a). We then analyzed neutrophils in embryos under germ-free condition and found that neutrophils accumulate in the digestive tract of mutant embryos (Fig. 4b), similar to that under standard culture condition (Fig. 3c). Thus, gut microorganisms are not a prerequisite for the inflammation in PIK3C3 mutants. We then assayed whether the neutrophil accumulation and inflammatory response has effect on the gut colonization of bacteria in mutants under standard culture condition. We determined the number of gut bacteria[20] from embryos at 6–8 dpf and found that they are similar in wild type siblings and mutants at 6 and 7 dpf (Fig. 4c). However, the number of gut bacteria drops from $2754 \pm 2632$ CFU/gut in WT ($N = 21$) to barely detectable in the mutant ($5 \pm 8$, $N = 14$) (Fig. 4c). These results indicate that gut bacteria are dispensable for the initiation of inflammatory response in the

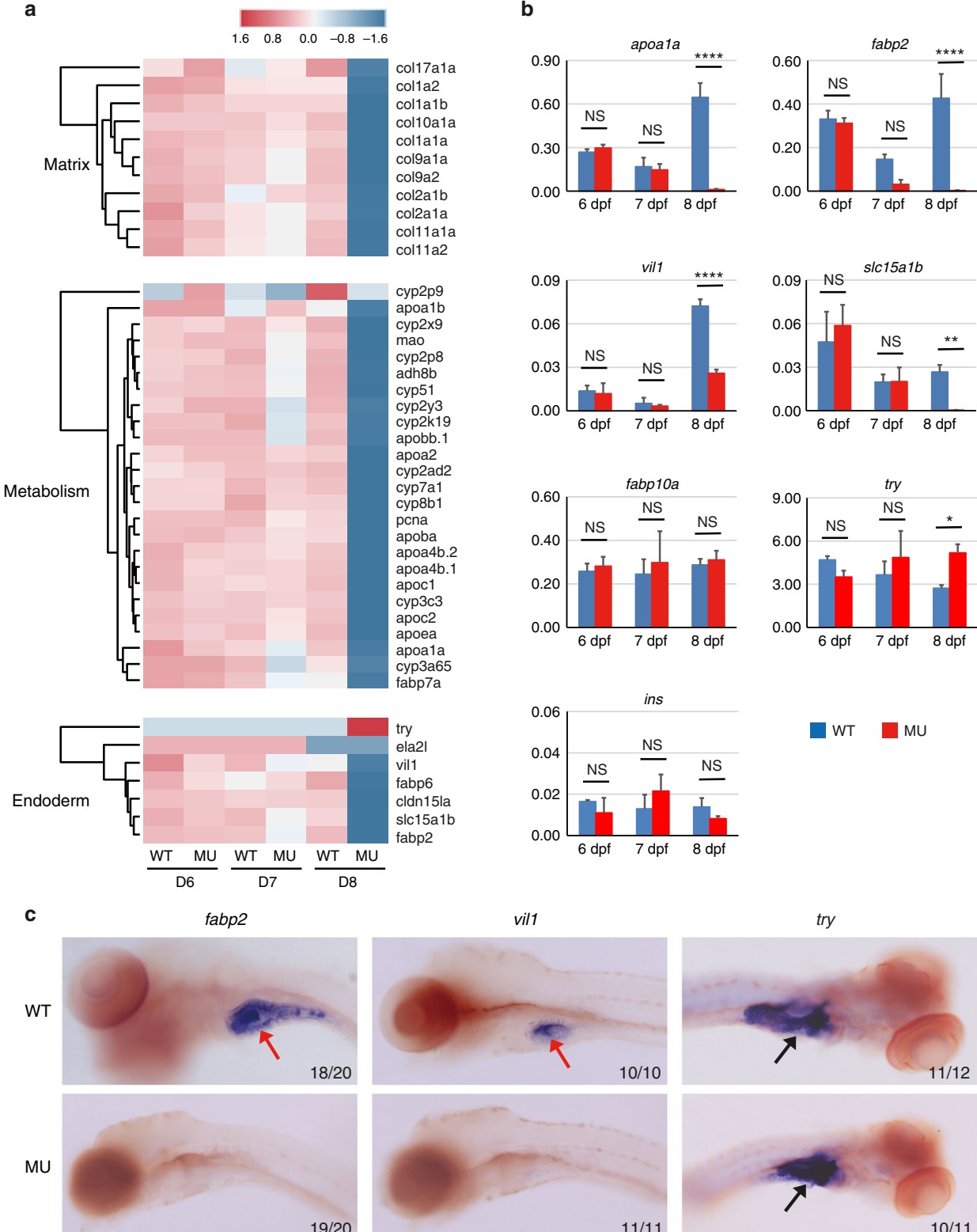

**Fig. 2** Defective gene expressions in *pik3c3* mutants. **a** RNA-Seq analysis of *pik3c3* mutants and control siblings at 6–8 dpf. Representative heatmaps show the expression of genes in one of these categories: extracellular matrix-related genes, liver and gut metabolic genes or markers of endoderm-derived cell lineages. **b** qRT-PCR analysis of the indicated genes from whole embryonic extracts. *apoa1a* is an apolipoprotein family member expressed in liver, gut, and other tissues. *fabp2* is intestinal fatty acid-binding protein. villin 1 (*vil1*) and *slc15a1b* (*pept1*) are markers of intestinal epithelium. *fabp10a* is liver fatty acid-binding protein. trypsin (*try*) and insulin (*ins*) are markers of exocrine and endocrine pancreas, respectively. *gapdh* is the internal control and data represent mean ± SD from three independent biological repeats (NS, non-significant; *$p < 0.05$; **$p < 0.01$; ****$p < 0.0001$ in one-way ANOVA with Tukey's multiple comparison). **c** In situ hybridization of 8 dpf embryos with the indicated probes. Red arrows point to intestines and black arrows indicate pancreases. Numbers represent (embryos with the indicated phenotype)/(total embryos analyzed)

mutants; on the other hand, the aberrant inflammation in *pik3c3* mutant gut functions to eliminate gut bacteria.

**PIK3C3 deficiency reduces PI3P level in IECs.** We next investigated how PIK3C3 regulates the homeostasis of intestinal epithelium. Class III together with class II PI3K are the enzymes that catalyze the synthesis of PI3P. We determined whether the level

and tissue distribution of PI3P are disrupted in the *pik3c3* mutants. 2xFYVE-GFP is a PI3P-specific probe[37] and we established a transgenic zebrafish line expressing 2xFYVE-GFP under the control of ubiquitously expressed ubiquitin promoter[38] to monitor the pattern of PI3P during early embryogenesis. We found that GFP signal is broadly distributed at early embryonic stages and it becomes enriched in somitic muscles starting from 2 dpf (Fig. 5a). PI3P is detectable in additional tissues, such as eye

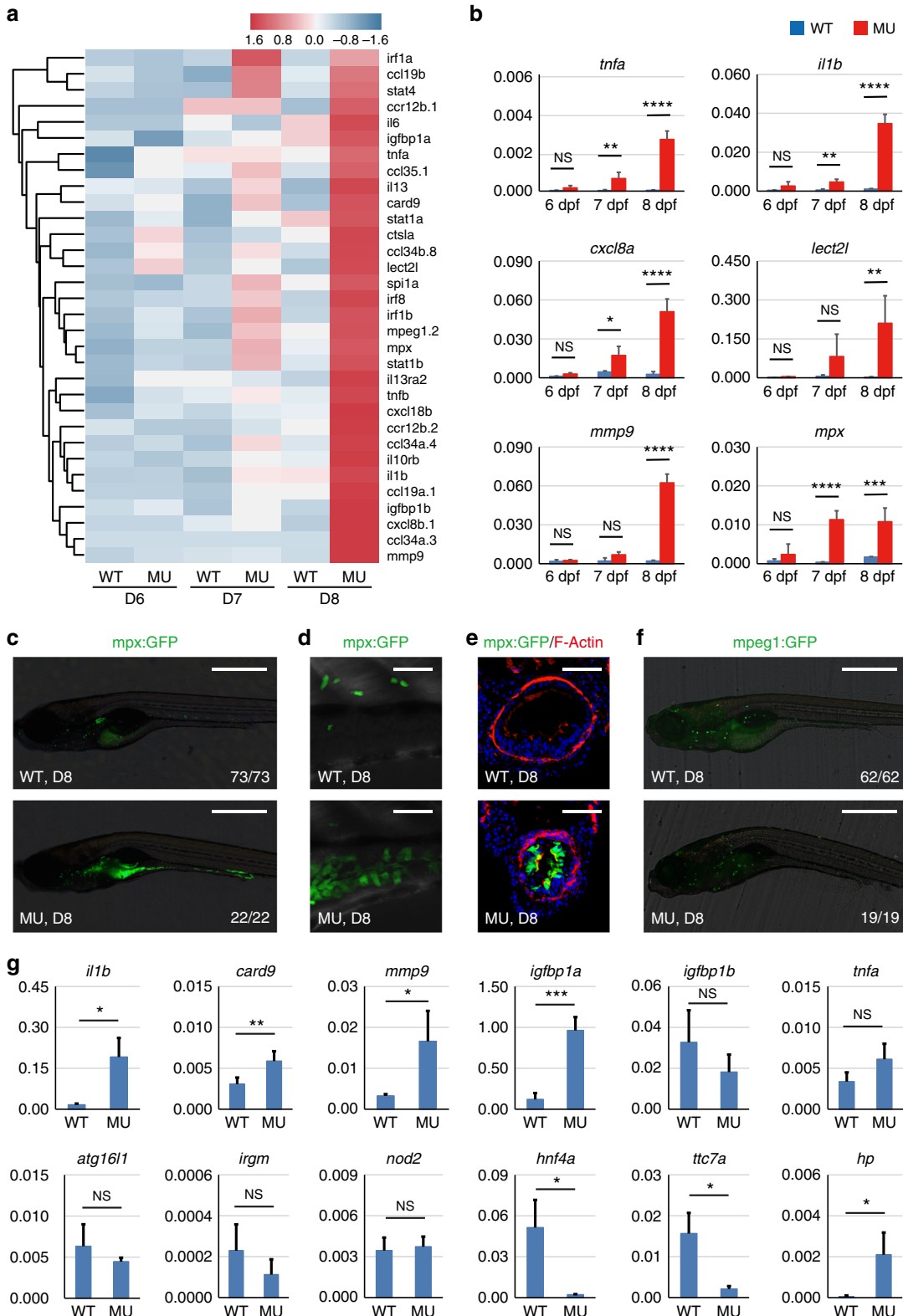

or pharyngeal arch at 6–8 dpf, indicating that the levels of PI3P are high in these tissues. We then determined the subcellular distribution of PI3P in these tissues. In cross sections of somite and gut, GFP-positive vesicles are clearly observed in WT IECs (arrow in Fig. 5b). However, a weak and diffusely distributed GFP signal is observed in the mutant IECs at 6–8 dpf (Fig. 5b, d). On the other hand, PI3P in the mutant somitic muscle cells appear normal at 6 dpf. At 7–8 dpf, somitic PI3P level gradually becomes reduced in the mutant (Fig. 5c, d). These results indicate that PIK3C3 is the dominant enzyme for PI3P production in IECs, while additional PI3Ks (such as members of class II PI3Ks) contribute to PI3P synthesis in somitic muscles.

**PIK3C3 deficiency induces mis-localization of E-Cadherin in IECs.** Our TEM analysis revealed that the integrity of IECs fails to be maintained in the *pik3c3* mutant (Fig. 1c). PI3P is a lipid involved in endosomal trafficking of membrane proteins, thus a reduction of PI3P in IECs may disrupt the polarized distributions of plasma membrane proteins, such as those involved in cell junctions and result in failure of homeostasis in IECs. We performed immunofluorescence staining to examine the distribution of some of those proteins. We found that wild type IECs show typical columnar arrangement and the adherens junction protein E-Cadherin is laterally distributed. However, the columnar shape and sheeted distribution of IECs are totally disrupted in the mutants. E-Cadherin at cell surface is clearly reduced while large intracellular aggregates of E-Cadherin are observed in mutant IECs (Fig. 6a). The aggregates do not co-localize with the lysosome marker LAMP1, indeed, the total protein level of E-Cadherin in the dissected mutant gut is not dramatically changes (Fig. 6b). The mRNA level of *cdh1* appears normal in the mutant (Fig. 6c). Thus, PIK3C3 deficiency results in the mis-localization but not the expression of E-Cadherin. Similar mis-localization occurs to the Na, K-ATPase protein ATP1A1, which normally shows basolateral distribution in IECs (Fig. 6a). On the other hand, the tight junction component ZO-1 is removed from cell surface and degraded in the mutants (Fig. 6a, b).

The disruption of PI3P signaling in mutant IECs may be responsible for the mis-localization of E-Cadherin in these cells, however, the aberrant inflammation may also play important roles in gut damage in mutants. To distinguish between these two possibilities, we adopted an in vitro Matrigel-based 3D culture of Caco2 cells (an epithelial cell line derived from human colorectal adenocarcinoma) to test the function of PI3P signaling in IEC polarity in the absence of immune response. A recent study reported that inhibition of PIK3C3 by siRNA or small chemical inhibitor disrupts polarized distribution of aPKC and epithelial integrity in Caco2-derived cyst[39], so we used this system to test whether the polarized distribution of E-Cadherin is regulated in a similar manner. We found siPIK3C3 treatment efficiently reduces the mRNA level of PIK3C3 in Caco2 cells (Fig. 6d). Epithelial cysts still form in treated cells, however, E-Cadherin is

internalized in 42% of the epithelial cyst derived from siPIK3C3-treated Caco2 cells ($N = 259$), while the number is 8% in the control siRNA treatment ($N = 282$, $p = 0.005$) (Fig. 6e, f). Interestingly, PIK3C3 knockdown does not change the protein level of E-Cadherin in Caco2 cells (Fig. 6g), which is consistent with the zebrafish result (Fig. 6b). Thus, PIK3C3 is required for the polarized distribution of E-Cadherin both in vitro and in vivo.

The E-Cadherin aggregates in the mutant could be the result of defective vesicular trafficking or degradation pathways. PIK3C3 is well-established to be involved in autophagy and endocytic trafficking pathway. For example, when complexed with BECN1/ATG14, PIK3C3 regulates autophagy; on the other hand, the PIK3C/BECN1/UVRAG complex is involved in endocytic trafficking pathways. We determined whether the autophagy or the endocytic trafficking pathway is involved in the PIK3C3 regulation of E-Cadherin. Western blot analysis of the dissected gut indicates that the levels of LC3B-II (an autophagy marker) are comparable between WT and *pik3c3* mutants at 6–8 dpf (Fig. 7a), indicating pathways other than autophagy is responsible for polarity defects in IECs. In the Caco2 in vitro cystogenesis model, we found that knockdown of UVRAG but not ATG14 induces abnormal epithelial lumen formation and mis-localization of E-Cadherin (Fig. 7b–d). Together with a recent report[39], these studies reveal a conserved PIK3C3-regulated endocytic trafficking pathway in the control of epithelial integrity from *Drosophila* to zebrafish to human organoid.

It has been reported that deficiency in CDIPT, an enzyme responsible for the de novo synthesis of intracellular PI, induces ER stress in IECs which disrupts the barrier function and promotes gut inflammation in zebrafish[20]. PI is the lipid substrate for PIK3C3 and the *cdipt* and *pik3c3* mutants share similar gut defect. Does ER stress have a role in *pik3c3* mutant guts? We found that the ER stress marker HSPA5 is upregulated at mRNA and protein levels in *pik3c3* mutants (Fig. 7e, f). However, HSPA5 is highly induced in liver and pancreas while its expression in gut is limited to a few cells in *pik3c3* mutants, which is different from that in *cdipt* mutants[20] or the tunicamycin-treated embryos (Fig. 7f). Together, our data suggest that it is the endocytic trafficking defects of polarity protein, such as E-Cadherin, but not defective autophagy or ER stress that results in gut defects in *pik3c3* mutants.

## Discussion

Phosphorylated PI lipids are critical regulators of intracellular vesicular trafficking. PI is synthesized through de novo pathway in ER and it can be phosphorylated by various PI kinases at the 3-hydroxyl, 4-hydroxyl, or 5-hydroxyl group of the inositol ring. For example, class II and III PI3-kinase convert PI to PI3P, which is further converted into PI(3,5)P2 by PIKFYVE[40]. In this study, we report that *pik3c3* mutants share similar IEC defects and intestinal inflammation with *cdipt* mutants. However, ER stress is not prominent in the *pik3c3* mutant guts (Fig. 7f) and we did not

**Fig. 3** Pro-inflammatory responses in the intestine of *pik3c3* mutants. **a** Heatmap of inflammation-related genes shows the gradual stimulation of inflammatory response from 6 to 8 dpf in the mutants. **b** qRT-PCR analysis of the indicated inflammation-related genes. *cxcl8a* is the zebrafish homolog of *IL8*. Experiments were performed and data analyzed as described in Fig. 2b. **c** mpx:GFP is a transgenic zebrafish line with neutrophils labeled by GFP. In 8 dpf mutants, neutrophils mainly accumulate in the digestive tract. Scale bar, 500 μm. **d** High magnification confocal imaging of (**c**). Scale bar, 50 μm. **e** Cross sections of intestinal bulb in D8 embryos. Neutrophil infiltration of intestinal epithelium is clearly seen in the mutants. Scale bar, 50 μm. **f** mpeg1: GFP-labeled macrophage lineage cells appear normal in the mutants. Scale bar, 500 μm. **g** qRT-PCR analysis of the expression of IBD candidate genes in dissected intestine of *pik3c3* mutants at 8 dpf. Inflammatory response genes (*il1b, card9, mmp9*) and *igfbp1a* (one of the homologs of human *IGFBP1*, a gene related to the prognosis of CD) are significantly up-regulated in the mutants. Bacteria sensing and autophagy-related genes (*nod2, atg16l1,* and *irgm*) and *tnfa* are not stimulated. Barrier function-related genes *hnf4a* and *ttc7a* are severely down-regulated in the mutants while the *hp* gene, which functions to open up tight junctions, is up-regulated in the mutants. *actb1* is used as internal control and data represent mean ± SD from three biological repeats. *p*-values are determined by unpaired two-tailed Student's *t*-test (NS, non-significant; *$p < 0.05$; **$p < 0.01$; ***$p < 0.001$; ****$p < 0.0001$)

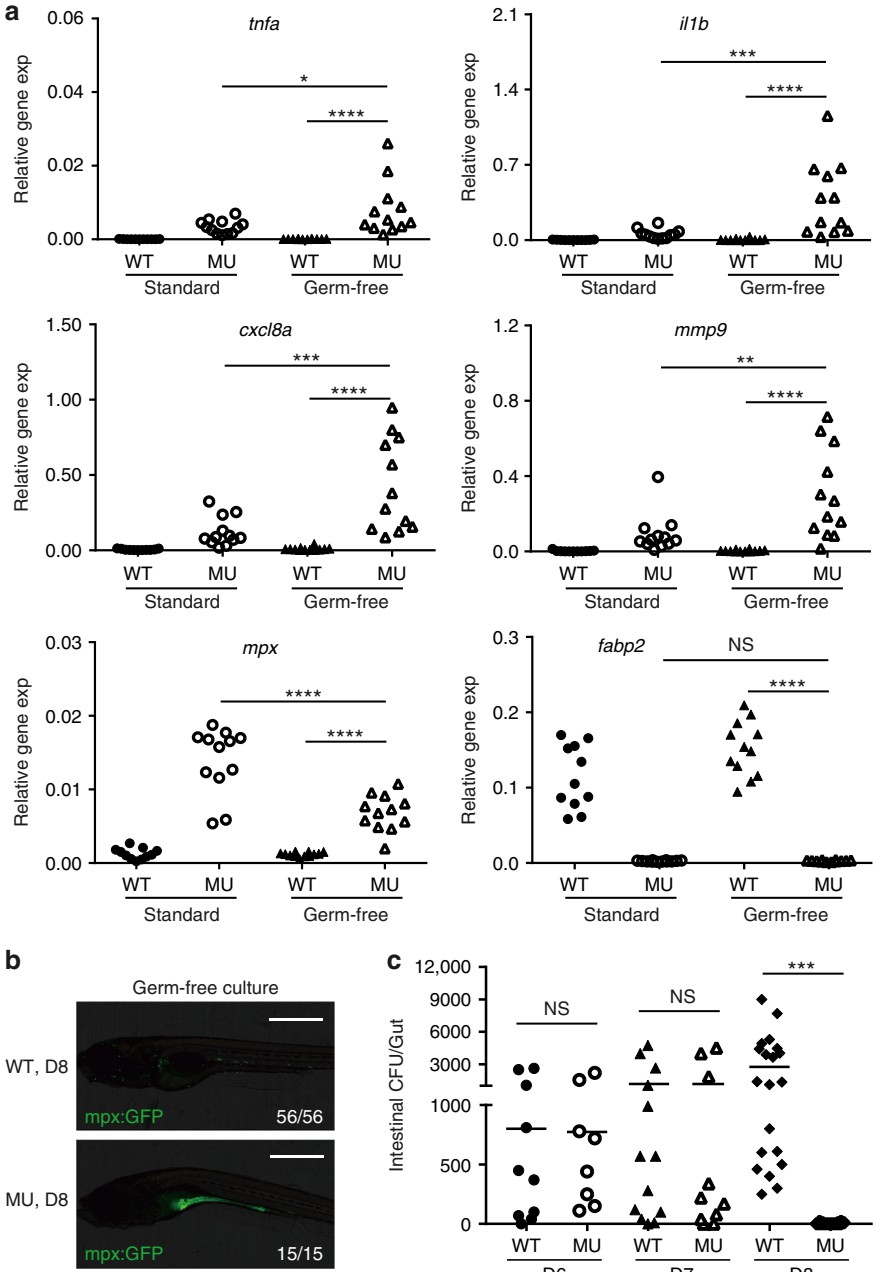

**Fig. 4** Intestinal microorganisms are not required for PIK3C3 deficiency-induced inflammation. **a** Scatter plots showing the qRT-PCR results of the indicated genes in 8 dpf embryos under either standard or germ-free culture conditions. Each dot represents the mean of three technical replicates from one embryo. Inflammatory genes are significantly stimulated in mutants under germ-free culture condition. NS, non-significant; *$p < 0.05$; **$p < 0.01$; ***$p < 0.001$; ****$p < 0.0001$ in one-way ANOVA with Tukey's multiple comparison. **b** Neutrophils accumulate in the digestive tract of mutant in the absence of gut microbiota. Images were acquired as described in Fig. 3c. **c** Scatter plot showing the numbers of bacteria in intestinal homogenates from individual embryo at 6, 7, and 8 dpf under standard culture condition. WT and mutants have similar gut bacteria at 6 and 7 dpf. However, the number of gut bacteria becomes either very low (<20 CFU/gut, 5 out of 14 embryos) or undetectable (9 out of 14) in the mutants at 8 dpf, which is significantly reduced when compared to that in wild type siblings (NS, non-significant; ***$p < 0.001$ in one-way ANOVA with Tukey's multiple comparison)

observe abnormal ER–Golgi structure in IECs of *pik3c3* mutants under TEM as reported in *cdipt* mutants. In addition, we notice that the number of gut bacteria is dramatically increased in *cdipt* mutants, while gut bacteria are eliminated in *pik3c3* mutants. Thus, there are similarities, as well as differences in these two mutants. Indeed, deficiency of PI is responsible for gut defect in *cdipt* mutant; on the other hand, PI (the substrate for PIK3C3) could accumulate in the absence of PIK3C3 (this possibility remains to be tested). It is possible that there are different underlying molecular mechanisms of gut defects in these two

mutants. Nevertheless, both studies indicate that a well-controlled PI metabolic pathway is essential for the integrity and function of IECs.

E-Cadherin is a major component of adherens junction in epithelium. Intestine-specific knockout of E-Cadherin induces loss of barrier function in mice. *CDH1* is a GWAS susceptibility gene for UC[6] and disease associated SNP in *CDH1* was reported in CD patients[7]. These studies indicate that the E-Cadherin mediate cell junction is essential for the integrity and function of intestinal epithelium. We notice that mis-localization, rather than

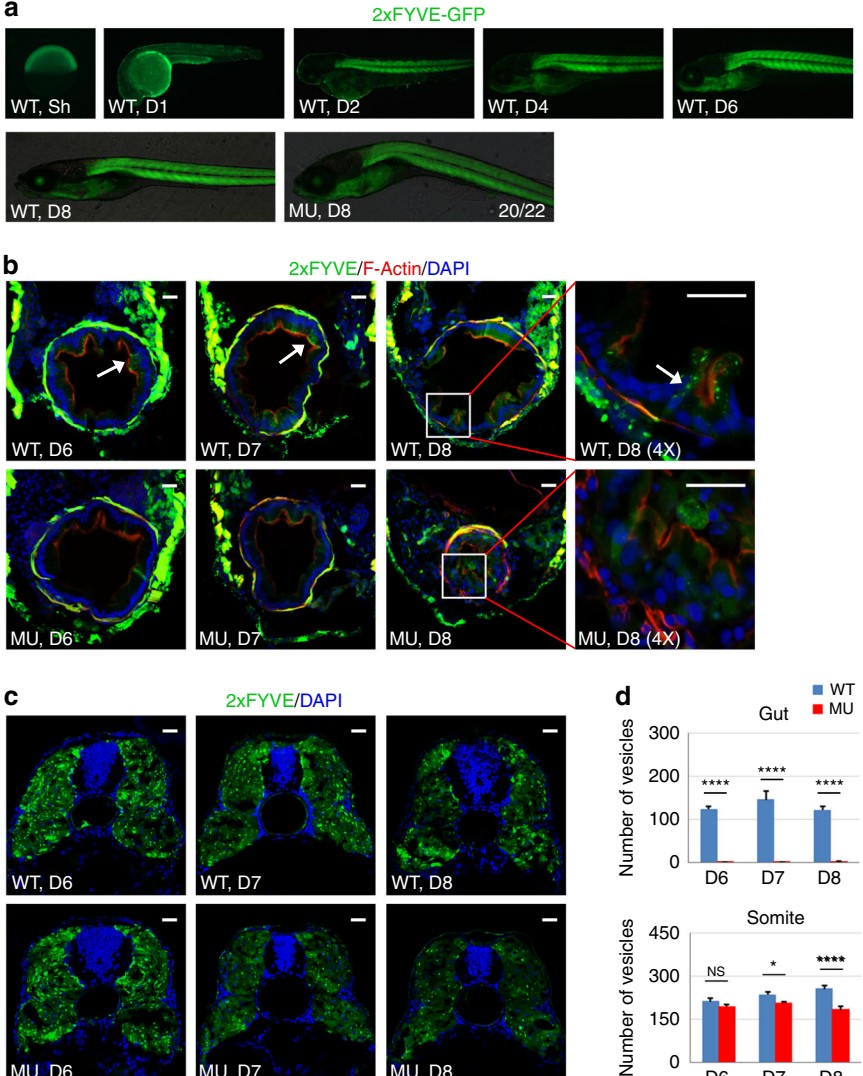

**Fig. 5** PI3P-positive vesicles are reduced in mutant IECs. **a** Represent images of transgenic zebrafish expressing the PI3P probe 2xFYVE-GFP under the control of ubiquitin promoter. Sh shield stage. **b** Cross sections of embryos at 6–8 dpf. PI3P-positive vesicles (arrow) are detectable in WT IECs, while the GFP signal in mutant IECs is weak and diffused. Scale bar, 20 μm. **c** PI3P-positive vesicles in somitic corss sections. Scale bar, 20 μm. **d** Quantification of PI3P-positive vesicles in (**b**, **c**). Data represent mean ± SD from three biological repeats (NS, non-significant; *$p < 0.05$; ****$p < 0.0001$ in one-way ANOVA with Tukey's multiple comparison)

degradation, of E-Cadherin is observed in several cases of gut defects. For example, intestine-specific knockout of *HNF4A* induces cytoplasmic retention of E-Cadherin[5]. Disease associated polymorphisms that result in cytoplasmic accumulation of E-Cadherin were reported in CD patients[7]. In this study, we report that deficiency of PIK3C3 and its lipid product PI3P induces similar cytoplasmic retention of E-Cadherin in zebrafish IECs, as well as in vitro cultured human Caco2 cells. These results indicate that post-translational regulations, especially vesicular trafficking pathways that recycle E-Cadherin to cell surface, play important roles in the proper distribution and function of E-Cadherin in IECs.

The vesicular trafficking of E-Cadherin has been extensively studied in epithelial cell line such as MDCK. It is well established that ubiquitinated E-cadherin is endocytosed and transported to lysosome for degradation via a Rab5 and Rab7 regulated pathway[41], or it is recycled back to basolateral membrane by a Rab11-mediated recycling pathway[42,43]. We recently reported that SNX16, a sorting nexin family member that preferentially binds to PI3P, is able to bind to E-Cadherin and function as an adaptor to promote the Rab11-dependent cell surface trafficking of E-Cadherin in renal epidermal cell line RCC10/VHL, as well as human embryonic stem cell line H1[44]. It is possible that SNX16 or another IEC-specific sorting nexin family protein functions in a similar manner to promote the recycling of E-Cadherin in IECs. A detailed characterization of this pathway is warranted since manipulation of this pathway might have therapeutic potential, especially in light of a recent discovery that a Rab11A-dependent pathway induced by yeast *Saccharomyces boulardii* CNCM I-745 supernatant is able to stimulate cell surface trafficking of E-Cadherin and restore intestinal barrier function in IBD patients[45].

## Methods

**Zebrafish lines**. Zebrafish stocks were maintained and handled according to standard protocols (zfin.org). Animal care and experimental protocols were approved by the Guangzhou Institutes of Biomedicine and Health Ethical Committee. The TU line zebrafish was used in the study. The transgenic lines *Tg(mpx: EGFP)* and *Tg2(mpeg1:EGFP)* were obtained from China Zebrafish Resource Center (CZRC). The *Tg(ubb:2xFYVE-EGFP)* was generated in house using the Tol2 transposon system. The PI3P probe 2xFYVE-EGFP (a gift from Dr. Alan R. Saltiel, University of Michigan) and a 3.5 kb *ubb* promoter[38] were cloned into the pT2A

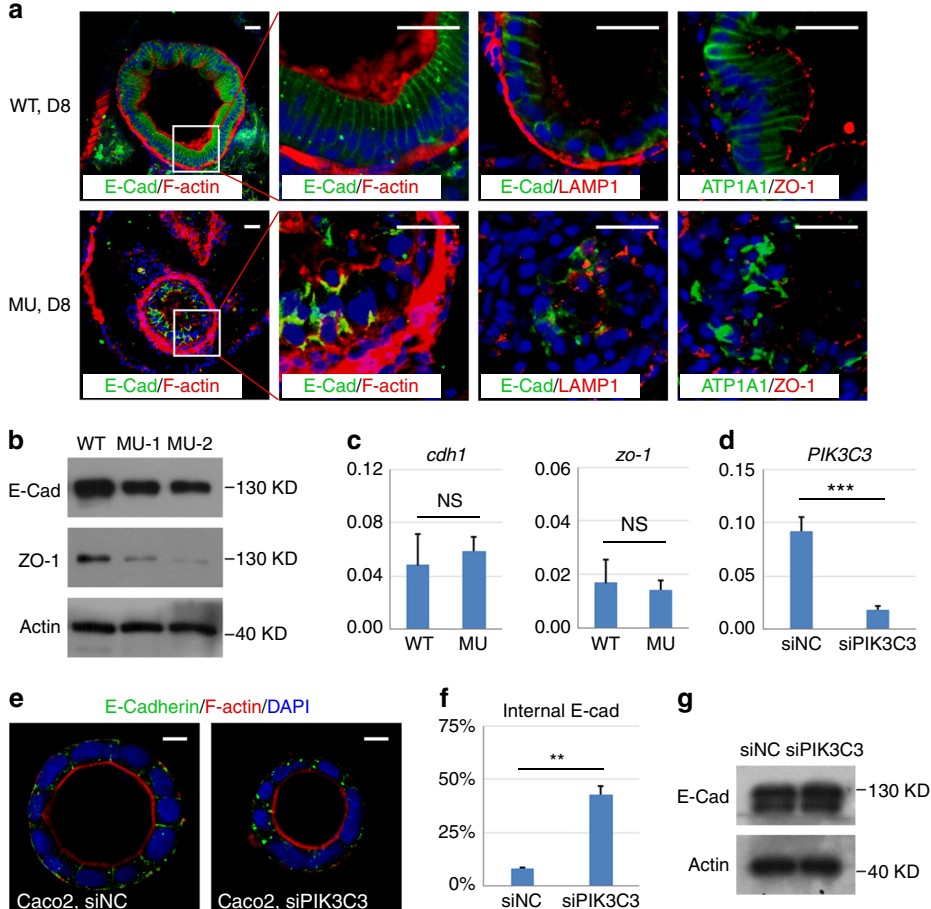

**Fig. 6** PIK3C3 deficiency induces polarity defects in IECs. **a** Immunofluorescence staining of gut cross sections at 8 dpf. In WT embryos, columnar IECs are arranged into a sheet and E-Cadherin are laterally distributed. IECs become disorganized and E-Cadherin is endocytosed and forms intracellular aggregates in the mutants. LAMP1 is a lysosome marker and it does not co-localize with E-Cadherin aggregates. Other polarity-related proteins including Na,K-ATPase alpha 1 subunit (ATP1A1) and tight junction protein ZO-1 also lose their polarized distributions in the mutant guts. Scale bar, 20 μm. **b** Western blot analysis for the expression levels of E-Cadherin and ZO-1 in the dissected digestive tracts from 8 dpf embryos. Beta actin is the loading control. **c** qRT-PCR analysis of *cdh1* and *zo-1*. Assays are performed as described in Fig. 3f. **d** Knockdown efficiency of *PIK3C3* by siRNA in Caco2 cells as determined by qRT-PCR. *ACTB* is the internal control and data represent mean ± SD from three biological repeats (***$p < 0.001$ by unpaired two-tailed Student's *t*-test). **e** PIK3C3 knockdown induces internalization of E-Cadherin in Caco2 cyst cultured on 3D Matrigel. Scale bar, 10 μm. **f** Quantification of (**e**). In each experiment, about 100 cysts were analyzed and data represent mean ± SD of the percentages of cyst with internalized E-Cadherin from three biological repeats (**$p < 0.01$ by unpaired two-tailed Student's *t*-test). **g** Western blot analysis of E-Cadherin protein levels in control or siPIK3C3-treated Caco2 cells. Beta actin is the loading control

vector. Detailed information about the construct is available upon request. Injection and F1 screen were carried out according to standard protocols. CRISPR/Cas9 system was used to knockout *pik3c3*. The targeting sequence 5′-GGCGACGGCATAGCGTCTGA-3′ was designed by Optimized CRISPR Design (http://crispr.mit.edu/). mRNA encoding a zebrafish-codon-optimized Cas9[46] (400 pg) together with gRNA (75 pg) were injected at one-cell stage. Two mutant alleles (−19 bp and −2 bp) were recovered from F1 and they induce same morphological and gut defects. The −19 bp deletion generates an EcoNI cutting site and it can be used for genotyping. For genotyping, DNA fragment was amplified with primers (5′-CAAGTGAAAGTTGAGTGCA TG-3′ and 5′-TCTCATTGCCAACGCTATT-3′) and subjected to sequencing or EcoNI digestion.

Germ-free experiments were carried out as described in ref. [36]. Briefly, fertilized eggs were washed three times with sterile E3 medium in EP tubes (less than 50 eggs per tube), then washed 5–10 times in antibiotics/E3 medium including 100 μg/mL ampicillin (A9518, Sigma), 5 μg/mL kanamycin (E004000, Sigma), 100 U/mL penicillin, and 100 μg/mL streptomycin (15140122, Thermo Fisher). They were transferred to sterile dishes with Pasteur pipettes and cultured in antibiotics/E3 medium. Medium were changed daily. Intestinal bacteria were tested at 8 dpf as described below.

**Intestinal bacteria analysis**. Zebrafish larvae at 8 dpf were anesthetized with tricaine (A5040, Sigma), washed three times in 0.1% Tween-20/PBS, and micro-dissected with sterile needles. Individual dissected intestine was washed in sterile

PBS for three times then homogenized in 50 μL PBS/0.25% trypsin (25200056, Thermo Fisher) with sterile pestles and incubated for 15 min at 37 ℃. Serial dilutions of the homogenates were plated on LB agar plates and incubated overnight at 28.5 ℃. The colony forming unit (CFU) for individual gut was calculated next day.

**Zebrafish digestive activity assay**. The in vivo assays for the digestive functions were performed with quenched fluorescent reporters[32]. PED6 (D23739, Thermo Fisher) and EnzChek (E6639, Thermo Fisher) were reconstituted according to manuals. Dextran (MW 4000, 46944, Sigma) was used to observe the swallowing activity. Larvae were immersed in E3 medium with 3 μg/mL PED6, 20 μg/mL EnzChek or 1% dextran for 3 h at RT, then imaged using the Zeiss Axio imager A1.

**Histological analysis**. Zebrafish larvae were fixed in 4% PFA/PBS overnight at 4 ℃, dehydrated in gradient ethanol and embedded in paraffin. 3 μm-transverse sections were prepared and stained with H&E according to standard protocol.

**Transmission electron microscopy (TEM) assay**. Zebrafish larvae were fixed in 4% (v/v) PFA and 2% (v/v) glutaraldehyde overnight at 4 ℃, washed in 0.1 M PBS (pH 7.4) and post-fixed in 1% OsO₄ for 1 h. Following gradient ethanol and propylene oxide dehydration, the samples were embedded in Epon 812 and polymerized for 48 h at 60 ℃. Cross sections (100 nm) corresponding to intestinal

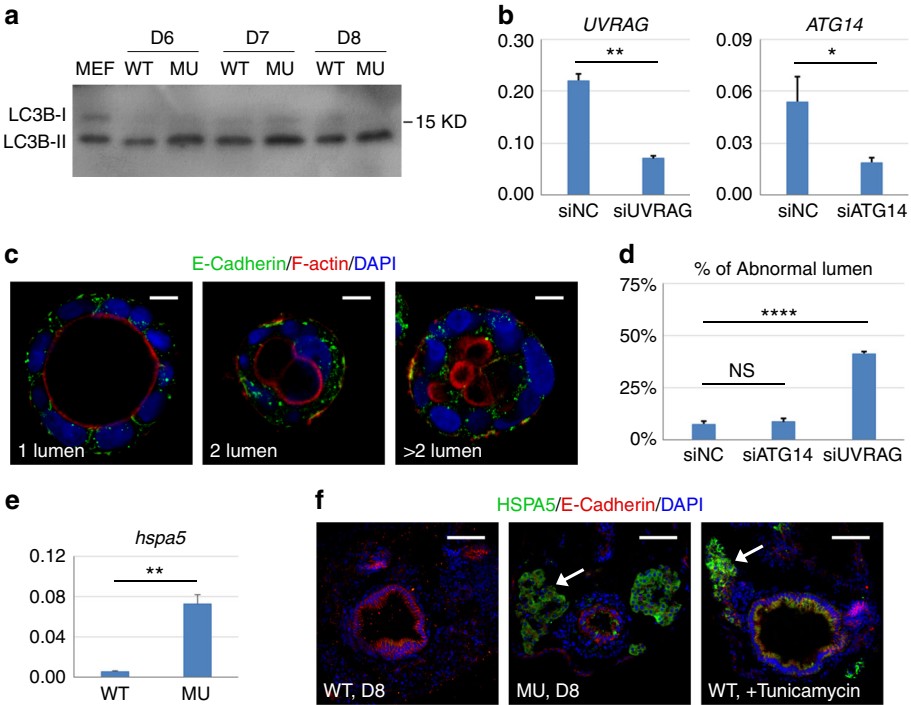

**Fig. 7** Defects in endocytic trafficking is responsible for PIK3C3 deficiency-induced polarity defect. **a** Western blot analysis of LC3B in dissected gut. MEF lysate (total cell lysate of mouse embryonic fibroblasts under autophagic conditions) was used as the positive control. The LC3B antibody can detect the activated form of zebrafish LC3B (LC3B-II) but not LC3B-I. **b** qRT-PCR analysis for the efficiency of siRNA to *UVRAG* or *ATG14* in Caco2 cells. *ACTB* is the internal control and data represent mean ± SD from three biological repeats (*$p < 0.05$; **$p < 0.01$ by unpaired two-tailed Student's *t*-test). **c, d** UVRAG but not ATG14 knockdown induces abnormal lumen formation and E-Cadherin mis-localization in Caco2 cyst. Scale bar, 10 μm. Data represent mean ± SD of the percentages of abnormal cyst (two or multiple lumens) from three biological repeats. NS, non-significant; ****$p < 0.0001$ by one-way ANOVA with Dunnett's multiple comparison. **e** qRT-PCR analysis of ER stress gene *hspa5* (heat shock protein family A member 5) in 8 dpf embryos. *gapdh* is the internal control and data represent mean ± SD from three independent repeats. *p*-value determined by unpaired two-tailed Student's *t*-test (**$p < 0.01$). **f** Immunofluorescence staining of gut cross sections. In the *pik3c3* mutant, HSPA5 signal is detected in liver (arrow) and a few cells in gut. Treatment of WT embryos with tunicamycin (an ER stress inducer, 1 μM from 7 to 8 dpf) strongly induces HSPA5 in liver and gut. Scale bar, 50 μm

bulb were prepared with ultramicrotome (EM UC7, Leica) and stained with uranyl acetate and lead citrate. Samples were then examined and photographed using the Tecnai G2 Spirit TEM (FEI).

**Quantitative real-time PCR (qRT-PCR).** Total RNA from whole embryo (either single embryo or group of 3–5 embryos) or dissected digestive tracts (from 10 embryos) was isolated using the TRI Reagent (TR 118, Molecular Research Center) according to the manual. cDNA was synthesized using the ReverTra Ace (TRT-101, TOYOBO). qRT-PCR was performed using the CFX96 real-time system (BIO-RAD) with SYBR Green Supermix (172-5274, BIO-RAD). Primers used in the study are listed in Supplementary Table 1.

**RNA-Seq and data processing.** Total RNA from 3 to 5 embryos at the indicated stages were prepared using the RNA Clean & Concentrator Kit (R1019, Zymo Research). cDNA library was constructed with the TruSeq RNA Sample Prep Kit (RS-122-2001, Illumina) and sequenced with the NextSeq 500/550 High Output Kit V2 (FC-404-2005, Illumina). Illumina bcl2fastq (v1.8.4) software was used for base calling. Sequenced reads were trimmed for adaptor sequence, and masked for low-complexity or low-quality sequence. The number of raw reads mapped to genes was calculated by RSEM (rsem-1.2.4), the reference genome using danRer10, then put all the sample results together and normalized by EDAseq (1.99.1), gene expression fold change was calculated using normalized raw reads. The downstream analysis was performed using glbase scripts. Reads were deposited with GEO under the accession number GSE102032.

**Whole-mount in situ hybridization.** Whole-mount in situ hybridization of 8 dpf embryos was performed according to standard protocol with minor modifications. Larvae were penetrated with 15 μg/mL proteinase K (03115852001, Roche) for 90 min at RT. Hybridization was performed for at least 16 h at 68 °C. Larvae were incubated with anti-digoxigenin-AP, Fab fragments (11093274910, Roche) overnight at 4 °C with gentle shaking followed by color reaction with NBT (11383213001, Roche)/BCIP (11383221001, Roche) for 30 min at RT. Pigment was removed with bleaching solution (0.5× SSC, 5% deionized formamide, and 10%

$H_2O_2$ in ddH$_2$O) for 15 min at RT. Probes used in this study include *fabp2*, *try*, and *vil1*. Detailed information about these probes is available upon request.

**Immunofluorescence staining.** Zebrafish larvae were fixed in 4% PFA/PBS overnight at 4 °C, washed three times in PBS and then transferred to embedding mold with tissue freezing medium (14020108926, Leica). Larvae were oriented vertically and cut into 10 μm using a cryotome (CM3050S, Leica). Sections were permeabilized with 0.5% Triton X-100/PBS for 30 min, blocked in 2% BSA/0.1% Tween-20/PBS for 1 h at RT and then incubated with primary antibody (diluted in blocking buffer, listed in Supplementary Table 2) overnight at 4 °C. For immunofluorescence staining of Caco2 cyst, cells were washed in PBS and fixed in 4% PFA at RT 30 min, then blocked in 3% BSA/0.2% Triton X-100/PBS and incubated with primary antibody overnight at 4 °C. After three washes with 0.1% Tween-20/PBS, smaples were incubated with a secondary antibody (1:50 in blocking buffer) for 2 h at RT. For F-actin staining, sections were incubated with 4 U/mL Alexa Fluor 568 Phalloidin (A12380, Thermo Fisher) for 2 h at RT. Samples were then washed and counter-stained with 1 μg/mL DAPI (D9542, Sigma) for 5 min. Mounted sections were imaged with the LSM 800 confocal microscope (Carl Zeiss).

**Western blot.** Caco2 cells or micro-dissected digestive tracts of 8 dpf larvae were collected in RIPA buffer (50 mM Tris–HCl pH 7.4, 150 mM NaCl, 1% NP40, 0.1% SDS) and SDS sample buffer (63 mM Tris–HCl pH 6.8, 10% glycerol, 5% β-mercaptoethanol, 3.5% SDS, and 0.01% bromophenol blue) containing protease inhibitor cocktail (04693132001, Roche) plus 1 mM PMSF (78830, Sigma). Samples were boiled for 10 min then centrifuged at 13,000×*g*. Supernatant was analyzed by Western blot according to standard protocol. Beta-actin was used as the loading control. Antibodies used in this study are listed in Supplementary Table 2. Uncropped blots are presented in Supplementary Figure 7.

**Tailfin amputation.** *Tg(mpx:EGFP)* larvae at 5 dpf were anesthetized and tail was amputated posterior to muscle and notochord with a sterile surgical blade and then cultured in a 24-well plate. Neutrophils recruitments were determined in a region of 100 × 500 μm from the wound site at 4 and 24 hours-post-amputation (hpa) as described in ref. [47].

**In vitro cystogenesis**. Caco2 cells (purchased from Shanghai Cell Resource Center) were cultured in DMEM plus 15% FBS, 100 U/mL penicillin, and 1 µg/mL streptomycin at 37 °C with 5% $CO_2$. This cell line was not contaminated by mycoplasma. For siRNA treatment, cells were seeded at $5 \times 10^4$ cells per well in a 24-well-plate and transfected with Lipofectamine RNAiMAX Transfection Reagent (13778150, Thermo Fisher) next day according to manufacturer's protocol. For cyst production, 5000 siRNA-transfected cells were suspended in 200 µl 5% Matrigel (354277, BD Biosciences) and then plated in eight-well chamber slide (154534PK, Thermo Fisher) pre-coated with 6 µL Matrigel. Medium were changed every other day and cysts were allowed to form in 5 days. The sequences of siRNAs are: siPIK3C3 (5′-ACUCAACACUGGCUAAUUAUU-3′)[39], siATG14 (5′-GCAAAU-CUUCGACGAUCCCAUAU-3′)[39], and siUVRAG (5′-UC ACUUGUGUAGUACUGAA-3′)[48].

**Statistical analyses**. Experiments were done from at least three biological repeats when possible. Data were presented as mean ± SD, which were calculated using Graphpad Prism 6 software (GraphPad). Statistical differences were determined by unpaired two-tailed Student's $t$-test between two groups, one-way ANOVA with Tukey's multiple comparison as a post hoc test for comparing every mean to every other mean or one-way ANOVA with Dunnett's multiple comparison as a post hoc test for comparing every mean to a control mean. $p$-value < 0.05 was considered statistically significant. No statistical method was used to predetermine sample size. No samples were excluded for any analysis.

**Data availability**. The RNA-Seq data generated in this study have been deposited to GEO under the accession no. GSE102032. All other relevant data are available from the corresponding authors upon request.

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

## Acknowledgements

We thank Dr. A. R.Saltiel (University of Michigan) for the PI3P probe 2xFYVE-GFP and Dr. X. Liu (GIBH) for reagents. The work was supported in part by Grants from the National Natural Science Foundation of China (31421004, 31530038), National Key R&D Program of China (2017YFC1001603, 2017YFA0504100), Key Research Program of Frontier Sciences of CAS (QYZDB-SSW-SMC031), and Guangdong Science and Technology Project (2014B050504008, 2014B050502012, 2017B030314056).

## Author contributions

X.S., D.P., and S.Z. designed research; S.Z., J.X., X. Wu., P.W., and Y.C. performed zebrafish studies; L.Z. and H.W. performed cell culture studies; H.L. performed TEM analysis; X. Wang performed bioinformatics; S.Z. and J.A. performed in vitro cystogenesis; X.S., D.P., S.Z. and Y.L. analyzed data; X.S. and S.Z. wrote the paper.

## Additional information

**Competing interests:** The authors declare no competing interests.

