## [Peer Review File · Nature Communications]

Reviewers' comments:

Reviewer #1 (Remarks to the Author):

The authors describe a novel zebrafish mutant for *pik3c3*, a gene encoding a class III PI3 kinase. This mutant shows clear and interesting gastrointestinal (GI) tract phenotypes, which is in line with the important role of phosphorylated forms of phosphatidylinositol in GI tract development and function. *Pik3c3* deficiency disrupts the distribution of cell-junction proteins in intestinal epithelial cells and results in GI tract inflammation.

The experiments are well-designed and clearly described in the manuscript. The data is convincing, but the comments listed below can hopefully help to further strengthen the message.

Minor comments:

- Figure 1A: The authors have generated a zebrafish mutant with a premature stop codon in the *pik3c3* gene. The truncation occurs before the crucial catalytic domain (PI3Kc), leaving the C2 domain intact and truncating the PI3Ka domain (Figure 1A). Is the mutant mRNA translated into a stable protein? Or is the aberrant mRNA, for instance, cleared via nonsense mediated decay?

Also, what is the function of the C2 and PI3ka domains and what would be the predicted remaining functionality of the truncated *Pik3c3* protein? In other words, is this a null mutant?

- Figure 1C: The mutants show a very severe GI tract phenotype at 8 dpf and onwards; do all mutants start to exhibit this phenotype or are there variations in severity?

Following up on that question, do heterozygous mutants show these phenotypes (as illustrated nicely with Dextran, PED6, and Enzchek) to a lesser extent?

- Figure 2B, 3B, and 4a: A ANOVA with post hoc test is more appropriate for the statistical analysis of this data.

- Lines 173 – 175 of the manuscript:

“Together, these results indicate that the digestive tract is the major site of inflammation in the mutants and this inflammation is likely mediated by neutrophils.”

The rise in inflammation clearly coincides with the decline in GI tract functionality and integrity. Do these GI tract phenotypes still occur if inflammation and/or neutrophil function is disrupted?

- Lines 191 – 193 and Figure 3F:

“We found that *nod2*, *atg16l1* and *irgm* genes are not induced in the mutants, indicating that the bacterial sensing and autophagy pathway are not affected in mutants.”

Nod2 is only one of many bacterial sensing proteins. Likewise, *Atg16l1* and *Irgm* are only two of many proteins involved in the autophagy pathway. It seems a bit of a stretch to extend these individual observations on gene expression levels to the functionality of two complex biological processes.

- Lines 217 – 218:

“Thus, it is the damage of IECs rather than abnormal gut microbiota that induces intestinal inflammation in the PIK3C3 mutants.”

Yes, this seems likely. However, there is a third option: aberrant inflammation damages IECs, which causes additional inflammation, etc. As mentioned before, it would be very informative to also follow mutant development in combination with disrupted immune or inflammatory functions

and assess whether damage to IECs still occurs at a similar level.

- Pik3c3 is part of two complexes with distinct functions: Pik3c3/Pik3r4/Becn1/ATG14 for autophagy and Pik3c3/Pik3r4/Becn1/UVRAG for endocytic trafficking. Since the inclusion of either ATG14 or UVRAG defines the function of the complex, knock down or knock out of these factors could confirm that proper E-Cadherin localisation indeed relies on Pik3c3-UVRAG dependent endocytic trafficking.

Yours sincerely,
Michiel van der Vaart, PhD

Reviewer #2 (Remarks to the Author):

The manuscript by Li et al. presents a novel zebrafish model for deficiency in the enzyme, pik3c3. The authors present a new mutant line, generated by CRISPR/Cas9, and the characterization of the gut phenotype, as well as examination of gene expression patterns and the impact of microbiota on gut inflammation. The model presented is an interesting one that will no doubt be useful, as demonstrated in this study. There are a number of controls, however, that are missing. These must be included in order to verify the validity of the model and the conclusions build on it:

1. The major area of validation that is missing is validation of disruption of the gene and protein in the mutant line. There is no demonstration of reduced RNA or protein. This should be included.
2. In addition, is the mutant phenotype dose-responsive? In other words, does the heterozygote also have a phenotype? This might offer insight into the mutation itself and also into the capacity for the system to withstand a reduction in the amount of the protein produced.
3. The gene expression studies carried out in intestinal tissue only are potentially very informative. However, no controls were included to ensure the purity of the intestinal tissue. Perhaps gene expression of intestine-specific transcripts could be performed in this sample and compared to another tissue sample, so as to ensure that the intestinal sample is indeed specifically or at least highly enriched in intestine. Moreover, expression of non-intestine genes should be assessed to ensure that there is not contamination from other tissues.
4. It is not clear how the germ-free fish are validated to be indeed free of microbiota. Can detection of gut microbiota - or lack thereof - be included so as to verify this?
5. The assessment of the number of gut microbiota and its impact on inflammation is very interesting. However, this experiment offers only a snap shot, rather than a longitudinal assessment of the impact of microbiota. Given that microbiome studies are characterized by extensive variation and that a snap shot may not accurately reflect the full story, can this be repeated at multiple time points during the first 10 days to demonstrate that there continues to be no impact from gut bacteria on the inflammatory response?
6. Finally, in Figure 5 the authors conclude that PI3P is reduced in the mutant IECs and this impacts other proteins. The reduction of PI3P has not been verified however. It should be demonstrated. Moreover, to validate the specificity of the effect, the authors can potentially reduce PI3P in a wild type system and examine whether the impact in IECs would be the same.

Minor points:

1. On line 141, what does "dominantly downregulated" mean?
2. There are multiple grammatical mistakes throughout the text.

3. Line 227: "wildly distributed" What does this mean?

Reviewer #3 (Remarks to the Author):

In this manuscript Zhao et al. have studied the effect of PIK3C3 (Vps34) deficiency in zebrafish. They find that in pik3c3 mutant intestine, the barrier epithelium is damaged and IBD-like inflammatory condition induced. The authors report that PIK3C3 deficiency results in intracellular redistribution of cell-junction proteins in the intestinal epithelial cells. Mechanistically this is suggested to be due to vesicular trafficking defects induced by PI(3)P deficiency. Previously (as authors also discuss in the paper), de novo PI synthesis defect has been reported to result to ER stress and inflammation in zebrafish intestine. As such, this paper seems to complement this finding, further underscoring the role of PI metabolism pathway in the development of IBD. The biggest drawback of the study is very meagre mechanistic dissection of the phenotype in relation to PI(3)P deficiency. The authors show that FYVE-probe, that binds to PI(3)P show less prominent punctate signal in pik3c3 mutant intestinal epithelium, indicative of reduced levels of the lipid in the IECs. In addition, they show that E-Cadherin and ZO-1 are removed from the cell surface, suggestive of vesicular trafficking defect. While this assumption is plausible owing to known roles of PI(3)P in vesicular trafficking, I feel that the data at present is insufficient to secure the conclusion.

Major comments:

1. CDIPT deficiency has previously been shown to result in ER stress and similar intestinal phenotype in zebrafish as PIK3C3 deletion. PI(3)P depletion may also result in defective autophagy, which can induce ER stress. ER stress has at least in keratinocytes been shown to result in intracellular redistribution of E-Cadherin and other junction proteins. I suggest that the authors study I) if the autophagic flux is normal and II) if ER stress is induced in their model.
2. I feel that the data describing PI(3)P deficiency in the enterocytes is insufficient. At the very least, the authors need to produce some quantifiable image data of the staining pattern, instead just showing pictures of two fish.
3. The same as point 2 holds true for the images showing E-Cadherin redistribution. While it is very clear that the distribution is altered, it would be beneficial to have larger magnification images of these cells and possibly co-staining of vesicular markers.

Minor comments:

1. Many abbreviated genes were not spelled out – at least those appearing in the text should be named.

Reviewers' comments:

Reviewer #1 (Remarks to the Author):

The authors describe a novel zebrafish mutant for *pik3c3*, a gene encoding a class III PI3 kinase. This mutant shows clear and interesting gastrointestinal (GI) tract phenotypes, which is in line with the important role of phosphorylated forms of phosphatidylinositol in GI tract development and function. *Pik3c3* deficiency disrupts the distribution of cell-junction proteins in intestinal epithelial cells and results in GI tract inflammation.

The experiments are well-designed and clearly described in the manuscript. The data is convincing, but the comments listed below can hopefully help to further strengthen the message.

Minor comments:

- Figure 1A: The authors have generated a zebrafish mutant with a premature stop codon in the *pik3c3* gene. The truncation occurs before the crucial catalytic domain (PI3Kc), leaving the C2 domain intact and truncating the PI3Ka domain (Figure 1A). Is the mutant mRNA translated into a stable protein? Or is the aberrant mRNA, for instance, cleared via nonsense mediated decay?

Our response: We now provide detailed characterization of the mutant in Supplementary Figure 1. qRT-PCR analysis reveals that the mutant RNA is stable (Supplementary Figure 1b), RT-PCR and DNA sequencing confirm that RNA splicing is normal in the mutants (Supplementary Figure 1c) and mutant RNAs have the same deletions as those in genomic sequences. We tried to detect the zebrafish PIK3C3 protein by western blot with at least 4 commercially available antibodies (Thermo Fisher #710194, Cell Signaling #4263, GeneTex #GTX129528 and Novus #NBP1-32545) but none of them worked, which is an unfortunate but common problem in zebrafish field.

Also, what is the function of the C2 and PI3ka domains and what would be the predicted remaining functionality of the truncated *Pik3c3* protein? In other words, is this a null mutant?

Our response: The C2 is a lipid binding domain involved in the membrane targeting of PIK3C3. PI3Ka is the accessory domain of PIK3C3, which could be involved in substrate presentation but remains to be confirmed. PI3Kc is the lipid kinase domain responsible for the production of PI(3)P. Both mutant alleles we generated result in premature truncation of PIK3C3 which miss 2/3 of the PI3Ka domain and the whole PI3Kc domain (Supplementary Figure 1a) and they are unlikely to have any lipid kinase activity. Will the truncated protein interferes with the wild type PIK3C3 protein function? We found that the heterozygous embryos have normal gut functions (Supplementary Figure 1e) and they grow up to adulthood with Mendelian ratio (Supplementary Figure 1d) and are fertile. Indeed, those heterozygous

animals are indistinguishable from their wild type siblings at all stages examined. Together, we conclude that the mutants we generated are null mutants.

- Figure 1C: The mutants show a very severe GI tract phenotype at 8 dpf and onwards; do all mutants start to exhibit this phenotype or are there variations in severity?

Our response: The phenotypes in both mutant alleles are very severe and synchronized. All embryos appear normal at 6 dpf and subtle morphological changes can be observed in mutants at 7 dpf. The mutants are clearly distinguishable from wild type embryos at 8 dpf (Fig. 1a, b) and no mutant survives to 10 dpf (Supplementary Figure 1d).

Following up on that question, do heterozygous mutants show these phenotypes (as illustrated nicely with Dextran, PED6, and Enzchek) to a lesser extent?

Our response: The heterozygous embryos are indistinguishable from their wild type siblings at all stages examined. The digestive activities in heterozygotes at 8 dpf are comparable to those in the wild type animals (Supplementary Figure 1e).

- Figure 2B, 3B, and 4a: A ANOVA with post hoc test is more appropriate for the statistical analysis of this data.

Our response: We appreciate the reviewer's comment and updated statistical analysis in all Figures and Supplementary Figures, including Figure 2B, 3B and 4A. A detailed description of statistical analysis is presented in the revised Methods section.

- Lines 173 – 175 of the manuscript: “Together, these results indicate that the digestive tract is the major site of inflammation in the mutants and this inflammation is likely mediated by neutrophils.”

The rise in inflammation clearly coincides with the decline in GI tract functionality and integrity. Do these GI tract phenotypes still occur if inflammation and/or neutrophil function is disrupted?

Our response: Aberrant inflammation is known to damage gut epithelia and we agree that it is critical to determine the roles of inflammation in PIK3C3 deficiency induced gut defects. Due to the relatively late gut phenotype in the mutants, we are not able to use the morpholino mediated loss-of-function studies to evaluate the function of inflammation in our model. We then turn to small chemical inhibitors. We treated control and *pik3c3* mutant embryos with anti-inflammation drug 5-ASA (1 mM, 6-8 dpf with daily change of media) or Caspase 1 inhibitor Ac-YVAD-CMK (80 μ M, 6-8 dpf, blocking IL1B activation) and a typical result is shown below. We found that neutrophils still accumulate in the mutant gut in the presence of above-mention small chemicals (a). Inflammatory genes (*tnfa*, *il1b*, *mmp9*) are not significantly blocked by those inhibitors either. The IEC marker *fabp2* is not recovered in the treated mutants. One caveat in this assay is that zebrafish embryos at 6-8 dpf may have

limited drug accessibility. We also tried injecting those drugs directly into circulation and obtained similar negative results.

a

b

Meanwhile, we used another strategy to investigate the function of PIK3C3 in epithelial integrity. When cultured in a Matrigel based 3D culture system, Caco2 cells (a human colorectal adenocarcinoma derived cell line) form epithelial cyst with polarity similar to gut. We found that PIK3C3 deficiency is able to induce internalization of E-Cadherin which is reminiscent of the situation in the mutant gut (Fig. 6a and 6f). Since there is no immune response in this in vitro cystogenesis model, we conclude that PIK3C3 is able to

cell-autonomously regulate intestinal epithelial polarity. Of course, PIK3C3 deficiency induced gut damage can lead to inflammation, which further damages gut epithelia.

• Lines 191 – 193 and Figure 3F:

“We found that *nod2*, *atg1611* and *irgm* genes are not induced in the mutants, indicating that the bacterial sensing and autophagy pathway are not affected in mutants.”

Nod2 is only one of many bacterial sensing proteins. Likewise, Atg1611 and Irgm are only two of many proteins involved in the autophagy pathway. It seems a bit of a stretch to extend these individual observations on gene expression levels to the functionality of two complex biological processes.

Our response: We agree with reviewer’s comment and removed the sentence and revised the text.

• Lines 217 – 218:

“Thus, it is the damage of IECs rather than abnormal gut microbiota that induces intestinal inflammation in the PIK3C3 mutants.”

Yes, this seems likely. However, there is a third option: aberrant inflammation damages IECs, which causes additional inflammation, etc. As mentioned before, it would be very informative to also follow mutant development in combination with disrupted immune or inflammatory functions and assess whether damage to IECs still occurs at a similar level.

Our response: The relationship between inflammation and IEC damage in vivo could be two-way interaction. We performed additional experiments to further dissect these interactions. We now report that knockdown of *PIK3C3* disrupts epithelial polarity in Caco2 derived cyst (Fig. 6f), indicating that *PIK3C3* deficiency in IECs is able to initiate polarity defect in the absence of immune response. This is not to say that inflammation has no function in the gut damage in *pik3c3* mutants, on the contrary, aberrant inflammation is likely to enhance gut damage through a positive feedback mechanism. As discussed above, we are not able to provide direct evidence on the role of inflammation in *pik3c3* gut damage due to technical problems. We have revised the text to better present our results.

• *Pik3c3* is part of two complexes with distinct functions: *Pik3c3/Pik3r4/Becn1/ATG14* for autophagy and *Pik3c3/Pik3r4/Becn1/UVRAG* for endocytic trafficking. Since the inclusion of either *ATG14* or *UVRAG* defines the function of the complex, knock down or knock out of these factors could confirm that proper E-Cadherin localisation indeed relies on *Pik3c3-UVRAG* dependent endocytic trafficking.

Our response: We used the Caco2 in vitro cystogenesis system to determine which of the *PIK3C3* complexes is involved in the regulation of E-Cadherin and epithelial polarity as suggested by reviewer. We found that knockdown of *UVRAG* but not *ATG14* induces polarity defect in Caco2 cyst (Supplementary Figure 8c, d). Together with a recent report

(Reference 39), it is clear that the PIK3C3/UVRAG dependent endocytic trafficking pathway plays conserved regulatory role in epithelial polarity from *Drosophila* to mammals.

Reviewer #2 (Remarks to the Author):

The manuscript by Li et al. presents a novel zebrafish model for deficiency in the enzyme, pik3c3. The authors present a new mutant line, generated by CRISPR/Cas9, and the characterization of the gut phenotype, as well as examination of gene expression patterns and the impact of microbiota on gut inflammation. The model presented is an interesting one that will no doubt be useful, as demonstrated in this study. There are a number of controls, however, that are missing. These must be included in order to verify the validity of the model and the conclusions build on it:

1. The major area of validation that is missing is validation of disruption of the gene and protein in the mutant line. There is no demonstration of reduced RNA or protein. This should be included.

Our response: We provide the results of further characterization of the mutant in Supplementary Figure 1. qRT-PCR analysis reveals that the mutant RNA is stable (Supplementary Figure 1b). RT-PCR and DNA sequencing confirm that there is no abnormal alternative RNA splicing in the mutant (Supplementary Figure 1c) and mutant RNAs have the same deletions as those in genomic sequences. The deduced mutant protein sequences are described in Supplementary Figure 1a. Both of them miss part of the PI3Ka domain (supposed to be involved in substrate presentation) and lack the lipid kinase domain (PI3Kc) and are expected to be null mutants. We indeed tried to detect PIK3C3 protein by western blot. Unfortunately, none of the commercially available antibodies tested (Thermo Fisher #710194, Cell Signaling #4263, GeneTex #GTX129528 and Novus #NBPI-32545) worked in the assay. Lack of appropriate antibodies is a common problem in the zebrafish related researches.

2. In addition, is the mutant phenotype dose-responsive? In other words, does the heterozygote also have a phenotype? This might offer insight into the mutation itself and also into the capacity for the system to withstand a reduction in the amount of the protein produced.

Our response: The heterozygotes are indistinguishable from their wild type siblings at all stages examined. They survive to adulthood with the expected Mendelian ratio (Supplementary Figure 1d). The digestive activities in heterozygous mutants as 8 dpf are comparable to those in the wild type animals (Supplementary Figure 1e).

3. The gene expression studies carried out in intestinal tissue only are potentially very informative. However, no controls were included to ensure the purity of the intestinal tissue. Perhaps gene expression of intestine-specific transcripts could be performed in this sample and compared to another tissue sample, so as to ensure that the intestinal sample is indeed

specifically or at least highly enriched in intestine. Moreover, expression of non-intestine genes should be assessed to ensure that there is not contamination from other tissues.

Our response: We compared the expression levels of lineage genes between whole embryos and dissected gut tissues by qRT-PCR (Supplementary Figure 5). We found that the gut gene *fabp2* is enriched more than sixty fold in the dissected samples. On the other hand, neuronal or muscle genes such as *pax2a*, *cmhc2* and *myh3.1* are barely detectable in the dissected samples. These results confirm the effectiveness of the dissection protocol.

4. It is not clear how the germ-free fish are validated to be indeed free of microbiota. Can detection of gut microbiota - or lack thereof - be included so as to verify this?

Our response: We used the protocol reported in Reference 36 to generate and validate germ-free embryos. For validation, intestine was dissected from individual embryo, homogenized and plated on LB agar plates after serial dilutions. A typical result is shown in Supplementary Figure 6. The assay was routinely performed for all germ-free related studies.

5. The assessment of the number of gut microbiota and its impact on inflammation is very interesting. However, this experiment offers only a snap shot, rather than a longitudinal assessment of the impact of microbiota. Given that microbiome studies are characterized by extensive variation and that a snap shot may not accurately reflect the full story, can this be repeated at multiple time points during the first 10 days to demonstrate that there continues to be no impact from gut bacteria on the inflammatory response?

Our response: We determined the number of gut bacteria at multiple time points and the results are presented in the updated Fig. 4c. At 6 dpf, the average number of gut bacteria is 801 in wild type siblings and it is 774 in the mutants, indicating that bacteria colonize both wild type and mutant guts. At 7 dpf, these numbers increase to 1162 and 1121 in control and mutants respectively. At 8 dpf, the number of gut bacteria in wild type sibling reaches 2565 while the average bacteria number in mutants decreases to 8. Indeed, most of the mutants (9/12) are free of gut bacteria. These results indicate that gut bacteria are eliminated by aberrant inflammation in the mutant guts after 7 dpf. The mutants become severely deformed at 9 dpf and none of them survives to 10 dpf (Supplementary Figure 1d).

6. Finally, in Figure 5 the authors conclude that PI3P is reduced in the mutant IECs and this impacts other proteins. The reduction of PI3P has not been verified however. It should be demonstrated. Moreover, to validate the specificity of the effect, the authors can potentially reduce PI3P in a wild type system and examine whether the impact in IECs would be the same.

Our response: PI(3)P is an essential lipid in cells and it is generated by either class II PI-3 kinase (*pik3c2a,b,g* in zebrafish) or class III PI-3 kinase (*pik3c3* in zebrafish). Knockout of any one member of the class II PI-3 kinase in zebrafish does not result in gut defect as seen in the *pik3c3* mutants (our observation), indicating *pik3c3* is the major enzyme responsible for

PI(3)P production in gut while class II PI-3 kinase may function in other tissues. Biochemical quantification of PI(3)P generally requires $>10^6$ cells which cannot be met for zebrafish embryonic gut tissues. So we established the 2xFYVE-GFP transgenic fish line to evaluate the tissue-specific distribution of PI(3)P. We performed fluorescence microscopy to semi-quantify PI(3)P in embryonic guts as well as somites from 6-8 dpf and the results are presented in the updated Fig. 5. We found that PI(3)P positive vesicles are dramatically reduced in mutant guts at 6-8 dpf. However, somitic PI(3)P vesicles are not reduced in the mutants at 6 dpf but gradually become reduced at 7-8 dpf. These results indicate that PIK3C3 deficiency has a stronger effect on PI(3)P production in gut than somitic muscles which is consistent with the mutant phenotype.

We have tried to inhibit PIK3C3 with small chemical inhibitor SAR405 by either adding it in the culture media (70 μ M, 5-8 dpf) or injecting it into circulation (1 mM, 6-8 dpf). However, we did not observe reduction of PI(3)P or gut defect in the 2xFYVE-GFP line. It could be due to poor bioavailability of the inhibitor in embryos at these stages.

Minor points:

1. On line 141, what does "dominantly downregulated" mean?
2. There are multiple grammatical mistakes throughout the text.
3. Line 227: "wildly distributed" What does this mean?

Our response: We have extensively revised the manuscript, corrected typos and mistakes in the above-mentioned sentences and other places as well. We believe that the revised manuscript is substantially improved both in data quality and writing.

Reviewer #3 (Remarks to the Author):

In this manuscript Zhao et al. have studied the effect of PIK3C3 (Vps34) deficiency in zebrafish. They find that in *pik3c3* mutant intestine, the barrier epithelium is damaged and IBD-like inflammatory condition induced. The authors report that PIK3C3 deficiency results in intracellular redistribution of cell-junction proteins in the intestinal epithelial cells. Mechanistically this is suggested to be due to vesicular trafficking defects induced by PI(3)P deficiency. Previously (as authors also discuss in the paper), de novo PI synthesis defect has been reported to result to ER stress and inflammation in zebrafish intestine. As such, this paper seems to complement this finding, further underscoring the role of PI metabolism pathway in the development of IBD.

The biggest drawback of the study is very meagre mechanistic dissection of the phenotype in relation to PI(3)P deficiency. The authors show that FYVE-probe, that binds to PI(3)P show less prominent punctate signal in *pik3c3* mutant intestinal epithelium, indicative of reduced levels of the lipid in the IECs. In addition, they show that E-Cadherin and ZO-1 are removed from the cell surface, suggestive of vesicular trafficking defect. While this assumption is

plausible owing to known roles of PI(3)P in vesicular trafficking, I feel that the data at present is insufficient to secure the conclusion.

Major comments:

1. CDIPT deficiency has previously been shown to result in ER stress and similar intestinal phenotype in zebrafish as PIK3C3 deletion. PI(3)P depletion may also result in defective autophagy, which can induce ER stress. ER stress has at least in keratinocytes been shown to result in intracellular redistribution of E-Cadherin and other junction proteins. I suggest that the authors study I) if the autophagic flux is normal and II) if ER stress is induced in their model.

Our response: The pioneering studies on zebrafish *cdipt* mutant inspired us to pursue the current study. PIK3C3 and CDIPT are in the same biochemical pathway and loss-of-function of these genes induces similar gut inflammation and injury, indicating they might share common molecular mechanisms. However, there are differences between these two mutants. For example, the gut defects appear slightly late in *pik3c3* mutants than *cdipt* mutants. There is no apparent ER stress in *pik3c3* mutant IECs while cell junction defects appear more severe in *pik3c3* than *cdipt* mutant IECs (Fig. 1d). HSPA5 immunofluorescence staining confirmed that ER stress is prominent in liver but not IECs in *pik3c3* mutants (Supplementary Figure 7b). The number of gut bacteria is reduced in *pik3c3* mutant at 8 dpf (Fig. 4c) which is in contrast to that in *cdipt* mutants. In addition, we did not observe obvious liver or eye defects in *pik3c3* mutants as reported in *cdipt* mutants. It appears that in the *pik3c3* mutants, the spectrum of defect is limited while the severity of IEC defects is stronger (We never achieved amelioration of gut defects in *pik3c3* mutants by anti-inflammatory drug such as 5-ASA or Caspase 1 inhibitor. Please also see our response to question 6 from reviewer 1). Thus, *pik3c3* and *cdipt* mutants share common features as well as have differences. Nevertheless, both mutants indicate that a well-controlled phosphatidylinositol pathway is required for the formation and function of gut epithelia.

PIK3C3 is well characterized for its role in autophagy. Is the gut defects in *pik3c3* mutants related to disrupted autophagy pathway? We compared the levels of LC3B protein between wild type and *pik3c3* mutant embryos at 6-8 dpf by Western blot (Supplementary Figure 8a). The antibody can detect the activated form of LC3B (LC3B-II) but not LC3B-I from zebrafish gut lysate. We tried Bafilomycin treatment in these assay but cannot reach a consistent conclusion (could be attributed to bioavailability issues in 8 dpf embryos). The preliminary data indicate that LC3B activation occurs in the mutants at all stages examined. We then turned to the in vitro cystogenesis model to determine whether autophagic or endocytic trafficking pathway is involved in polarity. We found that knockdown of UVRAG but not ATG14 disrupts epithelial polarity in Caco2 derived cyst (Supplementary Figure 8c, d), which is consistent with a recent report (Reference 39). Together, these results suggest that it is the PIK3C3/UVRAG complex mediated endocytic trafficking pathway that plays important role in IEC integrity and polarity.

2. I feel that the data describing PI(3)P deficiency in the enterocytes is insufficient. At the very least, the authors need to produce some quantifiable image data of the staining pattern, instead just showing pictures of two fish.

Our response: We combine the 2xFYVE-GFP transgenic fish line and confocal fluorescence microscopy to semi-quantify the tissue distributions of PI(3)P at 6-8 dpf embryos. For each stage, sections from at least three individual embryos were imaged for statistical analysis and the result is now presented in Fig 5b-d. We found that PI(3)P positive vesicles are dramatically reduced in mutant guts at 6, 7 and 8 dpf. On the other hand, somitic PI(3)P vesicles are not reduced in the mutants at 6 dpf but gradually become reduced at 7-8 dpf. These results indicate that PIK3C3 deficiency has a stronger effect on PI(3)P production in gut than somitic muscles which is consistent with the mutant phenotype.

3. The same as point 2 holds true for the images showing E-Cadherin redistribution. While it is very clear that the distribution is altered, it would be beneficial to have larger magnification images of these cells and possibly co-staining of vesicular markers.

Our response: We agree that it is critical to determine the exact distribution of E-Cadherin in the mutant IECs by co-staining. Indeed, we have tried very hard to determine the co-localization of E-Cadherin with a panel of endosome related markers including EEA1 (BD Biosciences #610457), Rab5a (ABclonal #A1180), Rab7 (ABclonal #A0631 and Abcam #ab50533) and HRS (GeneTex #GTX101718). Unfortunately, none of them works on zebrafish tissue section. Rab11 (Thermo Fisher #71-5300) and LAMP1 (Abcam #ab24170) worked in the assay but it is clear that E-Cadherin does not co-localize with them, indicating that the internalized E-Cadherin is not on recycling endosomes or lysosomes. Due to the very limited supply of antibodies that have been verified in zebrafish studies, we are currently not able to determine the exact subcellular distribution of internalized E-Cadherin in mutant IECs.

Minor comments:

1. Many abbreviated genes were not spelled out – at least those appearing in the text should be named.

Our response: We have revised the manuscript to include the full name of genes as suggested by the reviewer.

Reviewers' comments:

Reviewer #1 (Remarks to the Author):

The resubmitted manuscript has improved significantly. The authors have addressed almost all of my previous comments. Specifically:

1. The authors have provided a detailed characterisation of the mutant alleles and have now also clearly described the phenotype of heterozygous mutants.
2. The statistical analyses of data presented in figure 2B, 3B, and 4A have been corrected.
3. The analyses of a possible causative role for the inflammatory response in intestinal barrier disruption yielded negative or inconclusive results. Nonetheless, the additional data obtained with the in vitro epithelial cyst model strengthen the conclusion that Pik3c3-deficiency disrupts intestinal epithelial polarity (and thereby barrier function) via a cell-autonomous mechanism.
4. The authors have confirmed the involvement of UVRAG, but not ATG14 in this process.

There are some issues that require further clarification/correction:

1. The transgenic lines marking macrophages and neutrophils are referred to as mpeg1-GFP and mpx-GFP. This is incorrect, as this is not GFP fused to a protein, but the mpeg1 and mpx promoters driving GFP expression.
2. Lines 308 – 313: The analysis of ER-stress (Suppl Fig 7A and B) is part of the discussion. These results should be described in the results section.
3. Lines 325 – 331: Dissecting the requirement of UVRAG versus ATG14 (Suppl Fig 8A-D). This data should also be described in the results section.
4. Figure 3C and 4B: the panels depicting the mpx:GFP signal in the gut of WT and especially pik3c3 mutants require higher resolution/magnification images. With the current images it is very hard to discern the cell size and morphology typical for neutrophils and the GFP signal comes across as an intense 'blob' in the intestinal bulb. Even when neutrophil cluster together, individual cells normally remain discernible (see for instance supplementary figure 4a of this manuscript for higher resolution images of mpx:GFP-positive cells)

Reviewer #2 (Remarks to the Author):

All concerns have been adequately addressed and the manuscript is much improved. The conclusions of the study are supported by the data, which are now significantly strengthened including appropriate statistical analyses.

Reviewer #3 (Remarks to the Author):

The authors have sufficiently addressed the questions I raised. I would, however, suggest that they add a mention of E-Cadherin not being localized to lysosomes/recycling endosomes, as the stainings with Lamp1 and Rab11 were apparently successful.

Reviewers' comments:

Reviewer #1 (Remarks to the Author):

The resubmitted manuscript has improved significantly. The authors have addressed almost all of my previous comments. Specifically:

1. The authors have provided a detailed characterisation of the mutant alleles and have now also clearly described the phenotype of heterozygous mutants.
2. The statistical analyses of data presented in figure 2B, 3B, and 4A have been corrected.
3. The analyses of a possible causative role for the inflammatory response in intestinal barrier disruption yielded negative or inconclusive results. Nonetheless, the additional data obtained with the in vitro epithelial cyst model strengthen the conclusion that Pik3c3-deficiency disrupts intestinal epithelial polarity (and thereby barrier function) via a cell-autonomous mechanism.
4. The authors have confirmed the involvement of UVRAG, but not ATG14 in this process.

Our response: We thank the reviewer for the positive comments.

There are some issues that require further clarification/correction:

1. The transgenic lines marking macrophages and neutrophils are referred to as mpeg1-GFP and mpx-GFP. This is incorrect, as this is not GFP fused to a protein, but the mpeg1 and mpx promoters driving GFP expression.

Our response: We thank the reviewer for pointing out this issue. We have revised the text and figures to correct this error.

2. Lines 308 – 313: The analysis of ER-stress (Suppl Fig 7A and B) is part of the discussion. These results should be described in the results section.

Our response: The ER stress results are presented in the updated results section and Fig. 7 (e, f) now.

3. Lines 325 – 331: Dissecting the requirement of UVRAG versus ATG14 (Suppl Fig 8A-D). This data should also be described in the results section.

Our response: The UVRAG versus ATG14 data are presented in the results section and Fig. 7 (a-d) as suggested.

4. Figure 3C and 4B: the panels depicting the mpX:GFP signal in the gut of WT and especially pik3c3 mutants require higher resolution/magnification images. With the current images it is very hard to discern the cell size and morphology typical for neutrophils and the GFP signal comes across as an intense 'blob' in the intestinal bulb. Even when neutrophil cluster together, individual cells normally remain discernible (see for instance supplementary figure 4a of this manuscript for higher resolution images of mpX:GFP-positive cells)

Our response: We provide high magnification confocal images for mpX:GFP in WT and mutant guts in the updated Fig. 3d. Individual neutrophil is discernible at this magnification. Under a stereo microscope, the heavily clustered neutrophils are typically seen as shown in Fig. 3c.

Reviewer #2 (Remarks to the Author):

All concerns have been adequately addressed and the manuscript is much improved. The conclusions of the study are supported by the data, which are now significantly strengthened including appropriate statistical analyses.

Our response: We thank the reviewer for the positive comments.

Reviewer #3 (Remarks to the Author):

The authors have sufficiently addressed the questions I raised. I would, however, suggest that they add a mention of E-Cadherin not being localized to lysosomes/recycling endosomes, as the stainings with Lamp1 and Rab11 were apparently successful.

Our response: We thank the reviewer for the positive comments. We now provide the E-Cadherin/LAMP1 staining results in the updated Fig. 6a.

REVIEWERS' COMMENTS:

Reviewer #1 (Remarks to the Author):

All my remaining comments have been addressed and the manuscript presents clear and interesting conclusions that are supported by the data.

Reviewer #3 (Remarks to the Author):

The authors have addressed all the points that I had and in my opinion, the manuscript is suitable for publication